# VARIANCE COVARIANCE REGULARIZATION ENFORCES PAIRWISE INDEPENDENCE IN SELF-SUPERVISED REPRESENTATIONS

## ABSTRACT

Self-Supervised Learning (SSL) methods such as VICReg, Barlow Twins or W-MSE avoid collapse of their joint embedding architectures by constraining or regularizing the covariance matrix of their projector's output. This study highlights important properties of such strategy, which we coin Variance-Covariance regularization (VCReg). More precisely, we show that *VCReg combined to a MLP projector enforces pairwise independence between the features of the learned representation*. This result emerges by bridging VCReg applied on the projector's output to kernel independence criteria applied on the projector's input. This provides the first theoretical motivations and explanations of MLP projectors in SSL. We empirically validate our findings where (i) we put in evidence which projector's characteristics favor pairwise independence, (ii) we use these findings to obtain nontrivial performance gains for VICReg, (iii) we demonstrate that the scope of VCReg goes beyond SSL by using it to solve Independent Component Analysis. We hope that our findings will support the adoption of VCReg in SSL and beyond.

## 1 INTRODUCTION

Self-Supervised Learning (SSL) via joint embedding architectures has risen to learn visual representations outperforming their supervised counterpart. This paradigm enforces similar embeddings for two augmented versions of the same sample, thus allowing an encoder $f$ to learn a representation for a given modality without labels. Importantly, $f$ could solve the learning task by predicting the same embedding for every input, a failure mode known as collapse. To avoid this, various mechanisms have been proposed hence the diversity of SSL methods (*e.g.*, Grill et al. (2020); Caron et al. (2020; 2021); Chen & He (2021)). Most of these are a composition $g \circ f$ of the encoder with a projector neural network $g$. Only $f$ is retained after training, in opposition to supervised training that never introduces $g$. The projector was proposed by Chen et al. (2020) and significantly improved the quality of the learned representation in terms of test accuracy on ImageNet and other downstream tasks. Although some works (Appalaraju et al., 2020; Bordes et al., 2022) provide empirical knowledge on the projector, none provide a theoretical analysis of MLP projectors in practical SSL (Jing et al., 2021; Huang et al., 2021; Wang & Isola, 2020; HaoChen et al., 2021; Tian et al., 2020; Wang & Liu, 2021; Cosentino et al., 2022).

This study sheds a new light on the role of the projector via the lens of Variance-Covariance Regularization (VCReg), a strategy introduced in recent SSL methods (Bardes et al., 2022; Zbontar et al., 2021; Ermolov et al., 2021) to cope with collapse by constraining or regularizing the covariance or cross-correlation of the projector $g$ output to be identity. More precisely, we demonstrate that *VC regularization of the projector's output precisely enforces pairwise independence between the components of the projector's input i.e. the encoder's output*, and connects this property to projector's characteristics such as width and depth. This provides the first theoretical motivations and explanations of MLP projector in SSL. Fully or partially pairwise independent representations are generally sought for, *e.g* to disentangle factors of variation (Li et al., 2019; Träuble et al., 2021). Our experimental analysis suggests that different levels of pairwise independence of the features in the representation emerge from a variety of SSL criteria along with mutual independence. However, as opposed to other frameworks, VCReg allows for theoretical study and explicit control of the learned independence amount. We prove and experimentally validate this property for random projectors, study how it

emerges in learned projectors, and use our results to obtain new and significant performance gains on VICReg over Bardes et al. (2022). We then ablate the SSL context and lean on our findings to show that VCReg of a SSL projector solves Independent Component Analysis (ICA). Beyond providing a novel theoretical understanding of the projector, we believe that this work also leads to a better understanding of VICReg. The scope of VCReg is not limited to SSL: our experiments on ICA open the way to other applications where some degree of independence is needed.

## 2 BACKGROUND

### 2.1 MEASURING STATISTICAL INDEPENDENCE USING KERNEL METHODS

Measuring the independence between two sets of realizations $\{\boldsymbol{X}_1^1, \ldots, \boldsymbol{X}_1^N\}, \{\boldsymbol{X}_2^1, \ldots, \boldsymbol{X}_2^N\}, \boldsymbol{X}_1^i \in \mathbb{R}^M, \boldsymbol{X}_2^i \in \mathbb{R}^M$ is a fundamental task that has a long history in statistics e.g. through the Mutual Information (MI) of the two random variables $X_1$ and $X_2$ from which those two sets are independently drawn from (Cover, 1999). Those variables are said independent if the realization of one does not affect the probability distribution of the other. Computing the MI in practice is known to be challenging (Goebel et al., 2005), which has led to considerable interest in using alternative criteria e.g. based on functions in Reproducing Kernel Hilbert Spaces (RKHS) (Bach & Jordan, 2002), a special case of what is known as functional covariance or correlation (Rényi, 1959). It consists in computing those statistics after nonlinear transformation (Leurgans et al., 1993) as in

$$\sup_{f_1 \in \mathcal{F}_1, f_2 \in \mathcal{F}_2} \mathrm{Corr}\langle f_1(X_1), f_2(X_2)\rangle, \tag{1}$$

where $f_1, f_2$ are constrained to lie within some designed functional space, and the Cov can be used instead of the Corr. If Eq. (1) is small enough, then $X_1$ and $X_2$ are independent in regard to the functional spaces $\mathcal{F}_1, \mathcal{F}_2$. For example, if $\mathcal{F}_1$ and $\mathcal{F}_2$ are unit balls in their respective vector spaces, then Eq. (1) is just the norm of the usual correlation/covariance operator (Mourier, 1953) which would be enough for independence under joint Gaussian distributions (Melnick & Tenenbein, 1982).

**HSIC and pairwise independence.** More recently, Gretton et al. (2005a) introduced a pairwise independence criterion known as the Hilbert-Schmidt Independence Criterion (HSIC) which can be estimated given empirical samples $\boldsymbol{X}_1$ and $\boldsymbol{X}_2 \in \mathbb{R}^{N \times M}$ as

$$\mathrm{HSIC}(\boldsymbol{X}_1, \boldsymbol{X}_2) := \widehat{\mathrm{HSIC}}(X_1, X_2) = \frac{1}{(N-1)^2} \mathrm{Tr}(\boldsymbol{K}_1 \boldsymbol{H} \boldsymbol{K}_2 \boldsymbol{H}), \tag{2}$$

with $\boldsymbol{H}$ the centering matrix $\boldsymbol{I} - \boldsymbol{1}\boldsymbol{1}^T \frac{1}{N}$, $(\boldsymbol{K}_1)_{i,j} = k_1(\boldsymbol{X}_1^i, \boldsymbol{X}_1^j)$ and $(\boldsymbol{K}_2)_{i,j} = k_2(\boldsymbol{X}_2^i, \boldsymbol{X}_2^j)$ the two kernel matrices of $\boldsymbol{X}_1$ and $\boldsymbol{X}_2$ respectively, and $k_1, k_2$ of $\mathcal{F}_1, \mathcal{F}_2$ universal kernels such as the Gaussian kernel (see Steinwart (2001); Micchelli et al. (2006) for other examples). Crucially, since

$$\mathrm{HSIC}(X_1, X_2) \geq \sup_{f_1 \in \mathcal{F}_1, f_2 \in \mathcal{F}_2} \mathrm{Cov}\langle f_1(X_1), g(X_2)\rangle, \tag{3}$$

HSIC can be used to test for (pairwise) independence as formalized below.

**Theorem 1** (Thm. 4 from (Gretton et al., 2005a))**.** $\mathrm{HSIC}(X_1, X_2) = 0$ *if and only if $X_1$ and $X_2$ are independent.*

Gretton et al. (2005a) also provide a statistical test for pairwise independence based on HSIC. Further quantities such as upper bounds on the MI can be found in a similar way, e.g. see Thm. 16 in Gretton et al. (2005b). In our experiments, we will rely on HSIC under the Gaussian kernel scaled by the median of the distribution of pairwise euclidean distances between samples.

**dHSIC and mutual independence.** Mutual independence of a set of $D$ $M$-dimensional random variables $X_1, \ldots, X_D$ is a stronger property than independence between all pairs of random variables in the set. To evaluate it, Pfister et al. (2018) introduce dHSIC, a multivariate extension of HSIC. In short, dHSIC measures the distance between mean embeddings $\mu$ under a RKHS $\mathcal{F}$ (Smola et al., 2007) of the product of distributions and the joint distribution:

$$\mathrm{dHSIC}(X_1, \ldots, X_D) := \|\mu(\mathbb{P}^{X_1} \otimes \cdots \otimes \mathbb{P}^{X_D}) - \mu(\mathbb{P}^{X_1, \ldots, X_D})\|_{\mathcal{F}}. \tag{4}$$

Similarly to HSIC, Pfister et al. (2018) establish the equivalence between dHSIC = 0 and mutual independence along with a statistical test. We provide an estimator of dHSIC given empirical samples $\boldsymbol{X}_1, \ldots, \boldsymbol{X}_D$ of the above random variables in $\mathbb{R}^{N \times M}$ each as well as implementations of HSIC and dHSIC in Appendix D.

**Complexities.** In what follows, we consider the $D$ features of a batch of representations $\boldsymbol{X} \in \mathbb{R}^{N \times D}$ as $D$ scalar random variables ($M = 1$) with $N$ samples each. The subsequent complexities of HSIC for one pair of features and dHSIC are $\mathcal{O}(N^2)$ and $\mathcal{O}(DN^2)$ respectively. Testing pairwise independence between all variables in a $D$-set with HSIC is $\mathcal{O}(D^2N^2)$ while dHSIC requires $N \geq 2D$ (Pfister et al., 2018). Since competitive visual representations typically have $D = 2048$, we resort to surrogates to estimate independence of the representations in practice. These surrogates are detailed in Section 5.

## 2.2 VARIANCE-COVARIANCE REGULARIZATION IN SELF-SUPERVISED LEARNING

**SSL with joint embeddings** learns visual representations by producing two different augmented views of an input batch of images $\boldsymbol{S} \in \mathbb{R}^{N \times W \times H}$, denoted by $\boldsymbol{S}_{\text{left}}$ and $\boldsymbol{S}_{\text{right}}$. Each view is fed to an encoder $f$, typically a ResNet50 (He et al., 2016), producing representations $\boldsymbol{X}_{\text{left}}$ and $\boldsymbol{X}_{\text{right}} \in \mathbb{R}^{N \times D}$ which are passed through a projector $g$ to output embeddings $\boldsymbol{Z}_{\text{left}}$ and $\boldsymbol{Z}_{\text{right}} \in \mathbb{R}^{N \times P}$. Finally, an invariance term encouraging $\boldsymbol{Z}_{\text{left}}$ and $\boldsymbol{Z}_{\text{right}}$ to be similar is applied. While most SSL methods require architectural or training strategies to avoid collapse (Grill et al., 2020; He et al., 2021), Bardes et al. (2022) and Zbontar et al. (2021) only require to modify the loss. After training, only $f$ is retained to be used in downstream tasks. We will denote the $(2N, P)$ matrix $\boldsymbol{Z}_{\text{total}} \triangleq [\boldsymbol{Z}_{\text{left}}^T, \boldsymbol{Z}_{\text{right}}^T]^T$.

**VICReg.** In Bardes et al. (2022), an anti-collapse term $\mathcal{L}_{\text{VC}}$, which we coin VC regularization (VCReg), is added to an invariance loss to form $\mathcal{L}_{\text{VIC}}$:

$$\mathcal{L}_{\text{VC}} = \sum_{k=1}^{P} \max\left(0, 1 - \sqrt{\text{Cov}(\boldsymbol{Z}_{\text{total}})_{k,k}}\right) + \alpha \sum_{j=1, j \neq k}^{P} \text{Cov}(\boldsymbol{Z}_{\text{total}})_{k,j}^2, \qquad (5)$$

$$\mathcal{L}_{\text{VIC}} = \frac{1}{N} \sum_{n=1}^{N} \|(\boldsymbol{Z}_{\text{left}})_{n,.} - (\boldsymbol{Z}_{\text{right}})_{n,.}\|_2^2 + \mathcal{L}_{\text{VC}}, \qquad (6)$$

The leftmost term in $\mathcal{L}_{\text{VC}}$ corresponds to regularizing the variance of each feature in $\boldsymbol{Z}_{\text{total}}$ to be at least unit, while the second term seeks to minimize the covariance between each pair of features in $\boldsymbol{Z}_{\text{total}}$. Note that $\mathcal{L}_{\text{VC}}$ applies to each view separately. In Bardes et al. (2022), the weight of each term in $\mathcal{L}_{\text{VIC}}$ can be tuned. Since the authors find that best results are obtained with equal weights for the invariance and the variance terms, we only vary $\alpha$. Note that Bardes et al. (2022); Zbontar et al. (2021) observe that wider projectors further improve the representation learned by the encoder $f$, yet neither further study this intriguing phenomenon. Although this work focuses on VCReg as formulated in VICReg, we provide background on Barlow Twins and W-MSE in the Appendix A, while next section establishes the similarity between VICReg, Barlow Twins and W-MSE as VCReg optimizers.

# 3 VC REGULARIZATION OF SSL PROJECTOR'S OUTPUT ENFORCES PAIRWISE INDEPENDENT FEATURES AT THE ENCODER'S OUTPUT

In this section, we demonstrate that for random Multi-Layer Perceptrons (MLPs) projectors, minimizing VC of the projector's output amounts to minimizing HSIC —a measure of pairwise dependence (see Section 2.1)— between all pairs of features in the learned representation, *i.e* the projector's input. We then justify how this reasoning extends to learned projectors. Our claims are experimentally verified in Section 5.

**Notations.** In our setting, $\boldsymbol{X} \in \mathbb{R}^{N \times D}$ is the $D$ dimensional output of the encoder $f$ for a batch of size $N$. $\boldsymbol{X}$ is then fed to the projector $g$ to form embeddings $\boldsymbol{Z} = g(\boldsymbol{X}) = [g(\boldsymbol{X})_1, \ldots, g(\boldsymbol{X})_P] \in \mathbb{R}^{N \times P}$ on which SSL criteria are generally applied. Note that $\boldsymbol{Z}$ would be $\boldsymbol{Z}_{\text{left}}$ or $\boldsymbol{Z}_{\text{right}}$ in the previous subsection. The acronym MLP refers to projectors typically used in SSL, a neural network with three layers of same width, unless stated otherwise.

Our proof strategy consists in proving that VCReg of each MLP layer (Linear + Batch Normalization (BN) + ReLU) output enforces pairwise independence of its input, before composing these results. We first study nonlinear elementwise projectors $g : \mathbb{R} \mapsto \mathbb{R}^L$, which belong to the wider class of DeepSets (Zaheer et al., 2017), and of which BN followed by ReLU can be seen as an instance. We denote the mapping of such projectors as $\boldsymbol{Z} = g(\boldsymbol{X}) = [g(\boldsymbol{X}_{:,1}), \ldots, g(\boldsymbol{X}_{:,P})]$.

**Lemma 1** (Nonlinear elementwise projectors minimize HSIC of their input). *Let $g : \mathbb{R} \mapsto \mathbb{R}^L$ be a nonlinear elementwise projector; then, minimizing the covariance of $\boldsymbol{Z}$ with respect to the encoder $f$ amounts to minimizing HSIC on all feature pairs in $\boldsymbol{X}$ with kernels $\boldsymbol{K}_i = g((\boldsymbol{X})_{:,i})g((\boldsymbol{X})_{:,i})^T$.*

*Proof.* Let us consider the $N$ values of the $i^{\text{th}}$ data feature $(\boldsymbol{X}_{:,i})$ as realizations of a random variable. Recalling Eq. (2), independence of two random variables can be estimated via HSIC. Considering the arbitrarily complicated network $g$ and $\boldsymbol{Z} = [g(\boldsymbol{X}_{:,1}), \ldots, g(\boldsymbol{X}_{:,D})] \in \mathbb{R}^{N \times DL}$, we have:

$$\mathrm{HSIC}((\boldsymbol{X})_{:,i}, (\boldsymbol{X})_{:,j}) = \frac{1}{(N-1)^2} \mathrm{Tr}(g((\boldsymbol{X})_{:,i})g((\boldsymbol{X})_{:,i})^T \boldsymbol{H} g((\boldsymbol{X})_{:,j})g((\boldsymbol{X})_{:,j})^T \boldsymbol{H})$$

$$= \frac{1}{(N-1)^2} \left\| g((\boldsymbol{X})_{:,i})^T \boldsymbol{H} g((\boldsymbol{X})_{:,j}) \right\|_F^2$$

$$= \left\| \mathrm{Cov}\left( g((\boldsymbol{X})_{:,i}), g((\boldsymbol{X})_{:,j}) \right) \right\|_F^2$$

$$= \left\| \mathrm{Cov}\left( \boldsymbol{Z} \right)_{1+iL:1+(i+1)L, 1+jl:1+(j+1)L} \right\|_F^2$$

$$\implies \sum_{i \neq j} \mathrm{HSIC}((\boldsymbol{X})_{:,i}, (\boldsymbol{X})_{:,j}) = \left\| \mathrm{Cov}\left( \boldsymbol{Z} \right) \odot \left( (1 - \boldsymbol{I}_D) \otimes \mathbf{1}_L \mathbf{1}_L^T \right) \right\|_F^2,$$

and, in the case $L = 1$ we have $\mathbf{1}_L \mathbf{1}_L^T = 1$ leading to $\sum_{i \neq j} \mathrm{HSIC}((\boldsymbol{X})_{:,i}, (\boldsymbol{X})_{:,j}) = \sum_{i \neq j} \mathrm{Cov}(\boldsymbol{Z})_{i,j}^2$, concluding the proof. $\square$

To rigorously obtain independence, the $\boldsymbol{K}_i$'s must be universal kernels. To satisfy this assumption, we may randomize Batch Normalization by drawing the mean and variance from some distribution. Combining this operation with a ReLU, we obtain random elementwise nonlinearities approaching random features (Rahimi & Recht, 2007) of a universal kernel (Sun et al., 2018). Increasing $L$ improves the approximation of such kernel (Chen & Phillips, 2017) i.e. the larger $L$, the better approximation of HSIC the covariance term in VICReg is; see Appendix D for the randomized Batch Normalization implementation, Figure 8 for its justification, and Section 5 for empirical validation.

**Remark 2** (Necessity of variance regularization). *Although the variance regularization term does not explicitly appear when minimizing HSIC on all pairs, it is necessary when optimizing $\boldsymbol{X}$ to prevent the degenerate solution of $\boldsymbol{X}$ being a constant, a common collapse mode of SSL.*

**Lemma 2** (Random linear projectors minimize HSIC of their input). *Let $g$ be a random linear projector with weights $\boldsymbol{W}$, and $\boldsymbol{X}$ has same variance for each column. Then, for large projectors, minimizing the covariance of $\boldsymbol{Z} = g(\boldsymbol{X}) = \boldsymbol{X}\boldsymbol{W}$ with respect to the encoder $f$ amounts to minimizing HSIC with a linear kernel for each pair of features in $\boldsymbol{X}$. (Proof in Appendix B.1.)*

Since the corresponding kernel in lemma 2 is linear, only decorrelation can be achieved for such projector's input. This however differs from PCA as we optimize over $\boldsymbol{X}$ and not $g$'s parameters ($\boldsymbol{W}$). Proving lemma 2 requires a projector with orthogonal weights, i.e. $\boldsymbol{W}^T \boldsymbol{W} = \boldsymbol{I}$, which gets more and more accurate with random weights $\mathcal{U}(1/\sqrt{D}, 1/\sqrt{D})$ as $P$ (the output dimension of $g$) increases. This follows since the central limit theorem states that the dot-product between two $P$-dimensional weight vectors tends to $0$ with rate $\mathcal{O}(1/\sqrt{P})$. Weight initialization in neural nets roughly follows $\mathcal{U}(1/\sqrt{D}, 1/\sqrt{D})$, which will be used to implement random projectors.

**Theorem 3** (MLP projectors with random weights enforce pairwise independence.). *Let us consider a MLP composed of alternating random linear layers and elementwise nonlinearities. Then, for large projectors, minimizing the variance and covariance of the output $\boldsymbol{Z}$ enforces pairwise independence between all pairs of features in the input $\boldsymbol{X}$.*

*Proof.* Let us consider the last block of such MLP, *i.e.* a fully-connected linear layer followed by an elementwise nonlinearity. According to lemma 1, applying VCReg to the MLP, *i.e.* at the output of the nonlinearity will enforce pairwise independence under corresponding kernel for the input of the nonlinearity, which is also the output of the last linear layer. If the latter is wide enough so that it can be considered orthogonal, Theorem 11 in Comon (1994) ensures that pairwise independence is preserved for the input of the layer. We can then recursively extend the result backward to the whole MLP. If the last MLP layer is a fully-connected linear, then lemma 2 applies and we go back to the preceding elementwise nonlinearity. $\square$

Fig. 6 in Appendix C.1 shows that each layer in the random MLP is recursively VC-regularized from VCReg being applied only at the projector's output. Following Theorem 3, we can expect that wider projectors are beneficial to enforce pairwise independence while adding layers or learning the projector is not necessary; see Section 5 for empirical validation.

**Extension to BarlowTwins and W-MSE, and generality of VCReg.** Our results focus on VCReg as formulated in VICReg but can in fact be extended to methods that constrain the covariance of $Z$ explicitly, namely BarlowTwins and W-MSE (Balestriero & LeCun, 2022). Indeed, the objective in W-MSE (Equations 3-4 in Ermolov et al. (2021)) is VICReg with explicit constraint on the variance and covariance. Increasing the variance and covariance hyper-parameters in VICReg produces W-MSE hence our results extend seamlessly. The objective from Eq. (7) in BarlowTwins is also similar to VICReg. The derivation, deferred to Appendix B.2, shows that minimizing the constrained form of BarlowTwins objective from Eq. (7) is equivalent to minimizing VICReg's invariance term whilst explicitly constraining the variance covariance terms as in W-MSE; hence our results also hold for BarlowTwins; see Section 5 for empirical validation. In particular we will see that as BarlowTwins explicitly enforces minimum VCReg, it better optimizes HSIC compared to VICReg with standard hyper-parameters. Finally, as opposed to BarlowTwins loss and most SSL methods, VCReg can be used and be beneficial within single branch architectures. We provide such use case in Section 5.

**Learned projectors.** In state-of-the-art SSL representations, the projector is learned, which is not rigorously covered by Theorem 3. Complementing the study of Bordes et al. (2022), we argue that learning the projector is only crucial to satisfy the invariance criterion since random projectors are sufficient to obtain pairwise independent features, as demonstrated in Section 5. In fact, our experiments show that i) using VICReg, keeping the projector random generally produces lower HSIC of their input i.e. better enforces pairwise independence than learning the projector's weights, ii) using VICReg, learning the projector to optimize VCReg more strongly than the invariance term reduces performances, and iii) using VCReg, learning the projector, e.g. in the later studied ICA setting, creates a degenerate representation that does not enforce pairwise independence. We thus conjecture that learning the projector's parameters to mainly minimize VICReg's invariance term leaves the parameters close enough to their random initialization (Jacot et al., 2018) to maintain an accurate estimate of HSIC. In fact, we will see that the wider the projector, the less far away from initialization the parameters have to move, and the better optimized HSIC.

## 4 RELATED WORK

**Feature decorrelation.** Also known as whitening, feature decorrelation ensures that correlation between each pair of different features in a batch of feature vectors is zero, and that each feature has unit variance. It was originally used as a data pre-processing technique, see e.g. Hyvärinen & Oja (2000), before being extended to deep networks (Cogswell et al., 2015) as a regularizer. In the context of SSL, Hua et al. (2021); Ermolov et al. (2021) find that feature decorrelation helps solving the collapse issue. The former avoids collapse via appropriate Batch Normalization and its decorrelating variant (Huang et al., 2018) in the projector. This variant of Batch Normalization can be seen as the hard constraint counterpart of BarlowTwins. Practically, whitening can be implemented by learning a fully-connected layer (Husain & Bober, 2019) recovering Principal Component Analysis; as mentioned earlier (Section 3), this differs from VCReg which keeps the layer's parameters random and optimizes its input.

**Independence criterion for learning features.** Enforcing mutual independence to learn a representation has been proposed for example by Schmidhuber (1992). More recently, and in the context of supervised learning, Chen et al. (2019) demonstrated improved training of ResNets by reducing the pairwise dependence in the features at each layer via a combination of Dropout and Batch Normalization. By opposition, VCReg is applied to the projector's output and common SSL projectors do not rely on Dropout; we also show that Batch Normalization is not necessary to reduce pairwise dependence.

**HSIC in supervised and self-supervised learning.** HSIC-based losses have been employed in supervised learning e.g. see Mooij et al. (2009); Greenfeld & Shalit (2020) to ensure independence between the residual errors and the labels of a task at hand. Kornblith et al. (2019) measure similarity between two neural representations via HSIC. In SSL, Tsai et al. (2021) showed that a modified

version of BarlowTwins loss maximizes HSIC under a linear kernel between the embeddings of two augmented versions of the same sample. In a similar fashion, Li et al. (2021) proposed a SSL framework based on maximizing the dependence between the embeddings of transformations of an image and the image identity via HSIC. Both works differ from ours, which connects VCReg to the minimization of HSIC between pairs of features in the representation.

## 5 Experiments

We first put in evidence pairwise independence properties emerging in learned visual representations, before validating and exploiting our theoretical findings. Finally, we ablate the SSL context and demonstrate that the composition of a SSL-like random projector with VCReg induces enough independence to perform ICA. Importantly, none of the experiments with VCReg use Dropout, and Batch Normalization is not used when the projector is random. Hence, pairwise independence cannot be attributed to those two techniques as opposed to Chen et al. (2019); Hua et al. (2021).

### 5.1 Pairwise independence emerges in most visual representations

**Setup.** In these experiments, we track two metrics for statistical independence of the components in the learned representation during ResNet50 training on ImageNet with popular SSL frameworks, as well as a supervised baseline. The first metric, HSIC (Gretton et al., 2005a), tracks pairwise independence. The second metric, dHSIC (Pfister et al., 2018), tracks mutual independence. As explained in Section 2.1, neither HSIC nor dHSIC scale to the full representation (2048 components). Therefore, we instead rely on proxys: for pairwise independence, we compute and average HSIC on all pairs of the first $n$ components of the representation. For mutual independence, we sample sets of $n$ components for which we display dHSIC. We set $n = 10$ and compute these statistics on the whole ImageNet validation set. We expect both metrics to decrease during the training, which would suggest decreasing dependence among the representations. Although statistical tests are available for both metrics, we track bare HSIC and dHSIC as they are continuous. However, we perform HSIC tests in this first serie of experiments so that the bare values can be linked to concrete portions of independent pairs. The width of the Gaussian kernel in HSIC is scaled for each representation (see 2.1) to allow for comparison between methods. See Appendix E for the detailed setup.

**Results.** Figure 1 shows that different learning frameworks implicitly optimize pairwise independence of the features in the representation throughout the training (left), and at some point mutual independence (right). As expected from Section 3, HSIC is continuously optimized in Barlow Twins and VICReg (it increases for DINO)[1], and Barlow Twins better enforces pairwise independence since the covariance matrix is constrained. Precise values along with HSIC independence tests at level $\alpha = 0.5$ and corresponding test accuracies can be found in Table 1. Lower HSIC values do not necessarily entail more successful tests: it is possible to allocate low HSIC to sufficiently many pairs while having an overall larger HSIC. For example, Supervised has less independent pairs in spite of an overall smaller HSIC. Table 1 also suggests that pairwise independence is not predictive of test accuracy. Although it seems to be a desirable property, it is not sufficient to yield a good representation: for example, DINO achieves better test accuracy than BarlowTwins while having higher HSIC. We study this trade-off in the next subsection. Although we observe decreasing dHSIC, *i.e.* improvement of mutual independence in the representations, all methods quickly reach a floor and do not improve further although it is lower for Barlow Twins and VICReg. Hence, we cannot conclude that this property is properly optimized and do not study it in subsection 5.2, but conduct experiments to better evaluate mutual independence obtained by VCReg in subsection 5.3.

### 5.2 Projector characteristics fostering or hurting pairwise independence

**Setup.** These experiments validate the results from Section 3 by studying which projector characteristics help or hurt HSIC optimization when learning visual representations. The setup is the same as above except that dHSIC is not used anymore and the projector may be replaced by projectors introduced in Section 3: an elementwise nonlinear projector from lemma 1, random linear projectors of various width from lemma 2, and a MLP taking the representation ($D = 2048$) as input, and outputting an embedding of size $P$, with two hidden layers of width $P$ and ReLUs. The MLP can be

---

[1]HSIC is continuously optimized for SimCLR as well: this could be explained in light of Garrido et al. (2022), which connect VICReg and SimCLR, and is omitted here for conciseness.

[2]As opposed to other methods here, DINO relies on multicrop, which partly explains the performance gap.

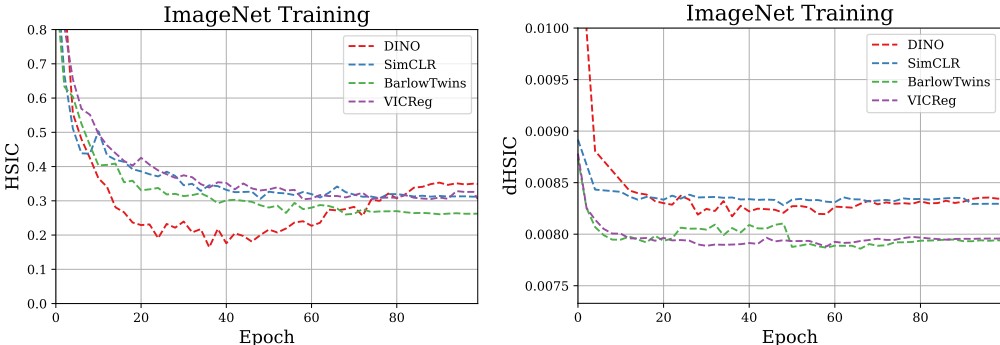

Figure 1: Overall, HSIC and dHSIC of the encoder output improves for all methods during training. VCReg based methods have smoother and/or further optimization of HSIC and dHSIC.

Table 1: Measure of pairwise dependence (HSIC), pairwise independence testing, and test accuracy for popular SSL or supervised representations averaged on multiple subsets of features.

| Method (100 ep.) | HSIC $\downarrow$ | % of indep. pairs $\uparrow$ | ImageNet Top1 $\uparrow$ |
|---|---|---|---|
| BarlowTwins (Zbontar et al., 2021) | $0.30 \pm 0.03$ | $38 \pm 2$ | 67.8 |
| VICReg (Bardes et al., 2022) | $0.39 \pm 0.03$ | $34 \pm 2$ | 68.4 |
| SimCLR (Chen et al., 2020) | $0.40 \pm 0.03$ | $36 \pm 1$ | 67.7 |
| DINO (Caron et al., 2021) | $0.38 \pm 0.02$ | $35 \pm 3$ | 70.4[2] |
| Supervised (Wightman et al., 2021) | $0.20 \pm 0.05$ | $33 \pm 3$ | 78.1 |

random as in Theorem 3 or learned as in Section 5.1. We apply an invariance loss along with VCReg on the output of the projectors and scale the covariance coefficient with the size of the projector for fair comparison. See Appendix E for the detailed setup.

**Results.** Figure 2 (top left) demonstrates that nonlinear elementwise projectors, random linear projectors and random MLPs with one layer achieve lower HSIC than learned MLPs, in particular when resampled so as to get better HSIC estimates. Hence, learning the projector is not necessary to obtain pairwise independence. As expected in Section 3, for both theoretical and classical projectors, increasing width yields lower HSIC while increasing depth hurts HSIC, see Figure 2. Table 5 in Appendix C.1 shows that increasing the weight of VCReg does not improve HSIC: a possible explanation is that the projector would otherwise start to optimize too much for VCReg. This intuition will be backed by Experiments 5.3. Modifying Batch Normalization to get closer to a universal kernel in lemma 1 yields significant test accuracy gains for VICReg over Bardes et al. (2022) (Figure 2, top right). Generally, low HSIC is beneficial provided that the test accuracy on ImageNet is reasonable: since HSIC does not guarantee that information is retained from the data, optimizing for it only results in performance loss, see inflexions in Figure 2 (top right) and Figure 5. We conjecture that HSIC combined to *e.g.* test accuracy on ImageNet could be used for model selection.

**Discussion.** As expected from Section 3, Batch Normalization in the projector is not required to enforce pairwise independence: Figure 2 (top right) shows that removing BN improves HSIC while maintaining test accuracy, which could be explained by the fact that each activation in the projector is implicitly regularized by VCReg (Figure 6), alleviating the need for BN. This could also explain why adding layers is detrimental to HSIC: implicit VCReg of the activations, an assumption of Theorem 3, will be more and more loosely enforced. Although projectors from Section 3 learn representations with better pairwise independence properties than classical SSL methods, most of them are not as competitive in terms of test accuracy (Fig. 2, top right). This complements the view of Appalaraju et al. (2020); Bordes et al. (2022), which argue that the projector requires some learning capacity to filter out information that is irrelevant to the invariance criterion. Our experiments suggest that the projector capacity should rather be increased via width than via depth: adding layers can be detrimental to HSIC as seen above but also to test accuracy (Appalaraju et al., 2020; Chen et al., 2021).

### 5.3 ABLATION: INDEPENDENT COMPONENT ANALYSIS (ICA) WITH VCREG

The following experiments aim to demonstrate that our findings hold outside of SSL. To this end, we show that VCReg of a SSL-like projector's output induces enough pairwise independence to solve

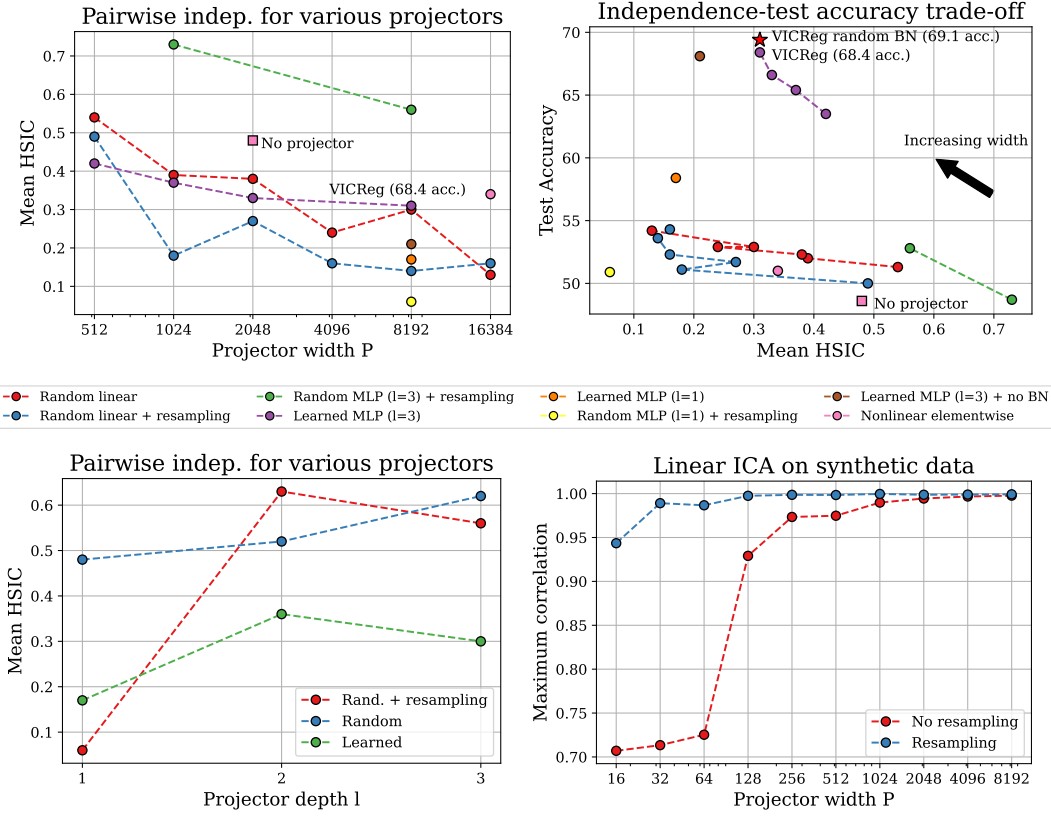

Figure 2: Top left: wider projectors and resampling both yield more pairwise independence for the encoder output. Bottom left: increasing depth hurts HSIC for random or learned projectors. Top right: trade-off between independence of the representation and test accuracy (width $D = 8192$). Bottom right: ICA experiments. Resampling and increased width improve the quality of the reconstruction.

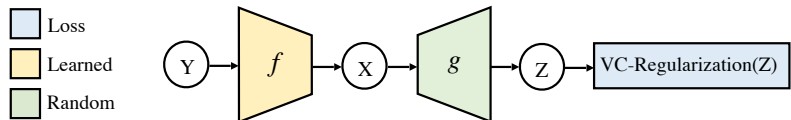

Figure 3: Linear ICA model expressed from VCReg of the projector's output.

linear ICA, *i.e.* recovering independent sources $S \in \mathbb{R}^{N \times D}$ from a mixture $Y = SA \in \mathbb{R}^{N \times D}$. See Appendix A for a primer on ICA.

**VC regularized projectors solve linear ICA.** In this setting, finding $M$ enforcing *pairwise independence* in the components of $YM$ is generally sufficient to recover $S$ (Comon (1994), Theorem 11). VCReg of a random projector's output should therefore be able to recover $S$. Our model can be seen on Figure 3: whitened batches of mixtures $Y$ are fed to an encoder $f$. Here, $f$ is a linear transformation $M$ described above. The output $X = f(Y)$ is then fed to a projector $g$, and VCReg is applied to the output $Z = g(X)$ covariance matrix. As in Experiments 5.2, the projector is randomly resampled at each gradient step to get better HSIC estimates. This setting corresponds to one branch of a VICReg network i) with a linear projection $M$ encoder instead of a neural network, ii) with a random projector, and iii) without invariance criterion. We perform ICA on two datasets, a synthetic one (Brakel & Bengio, 2017) with 6 sources among which 2 are noise, and a real audio dataset (Kabal, 2002) with 3 sources among which one is noise, also used in Brakel & Bengio (2017). The evaluation metric is the maximum correlation between the true and reconstructed sources. As this metric is not available without ground truth, we select the model with lowest dHSIC for $X$, which can be perfectly evaluated here since the number of sources is small. Our model recovers the sources, and is even competitive with methods specifically designed for linear ICA such as Fast

Table 2: Maximum correlation (between true and reconstructed sources (the higher the better ↑). The projector performs competitive reconstruction in the linear setting (left), where pairwise independence is sufficient, but not in PNL (right) where it is not.

| | Linear mixture | | | Post Non Linear mixture | |
|---|---|---|---|---|---|
| Method | Synthetic data | Audio data | Method | Synthetic data | Audio data |
| Whitening | 0.8074 | 0.9876 | Whitening | 0.7981 | 0.9046 |
| FastICA | 0.9998 | 1.0 | FastICA | 0.8311 | 0.8989 |
| Anica | 0.9987±6.5e-4 | 0.9996±4.9e-4 | Anica | 0.9794±53e-4 | 0.9929±18e-4 |
| VCReg | 0.9986±8.2e-4 | 0.9936±64e-4 | VCReg | 0.8465±142e-4 | 0.8706±376e-4 |

ICA (Hyvärinen & Oja, 2000) or Anica (Brakel & Bengio, 2017), see Table 2. Both increasing the width of the projector and resampling it at each step (especially for smaller projectors) improve the reconstruction as can be seen in Figure 2 (bottom right). This is in line with our findings in Section 3 and Experiment 5.2.

**VC regularized projectors do not solve nonlinear ICA.** Experiment 5.1 suggests that mutual independence also improves during training, although not as clearly as pairwise independence. Hence, one could ask whether VCReg also enforces the former enough to solve nonlinear ICA. To test this hypothesis, we apply VCReg to a particular case of nonlinear ICA which allows identifiability but does not have equivalence between pairwise and mutual independence: the post-nonlinear mixture (PNL) (Taleb & Jutten, 1999). In PNL, the sources are linearly mixed before being fed to elementwise nonlinear functions. For these experiments, and following Brakel & Bengio (2017), our encoder is a MLP. During our first experiments, we observed an informational collapse of the encoder, which produces seemingly mutually independent variables with very poor reconstruction of the sources. To alleviate this issue, we add a decoder taking $X$ as input and reconstructing $Y$. Figure 9 in Appendix E shows our modified setup. We compare VCReg to FastICA and Anica. Although FastICA is not meant to solve the nonlinear case, it remains an interesting baseline. Table 2 shows that our model fails to recover the sources, as it does only slightly better than FastICA in the synthetic case. Hence, although mutual independence increases during training, VCReg does not optimize it enough to solve nonlinear ICA. We propose a simple explanation for this limitation. Indeed, each feature in $Z$ is a nonlinear function of all features in $X$. Hence, it is not possible to completely decorrelate two components of $Z$ as they both contain the same set of features. It is still possible to improve mutual independence to some extent since, in practice, only parts of the inputs are considered at once by nonlinear mappings such as neural networks (Erhan et al., 2009; Adebayo et al., 2018).

**Learning the projector does not enforce pairwise independence.** ICA experiments done with a learned projector fail to recover independent sources: this further demonstrates that learning the projector to specifically optimize the VCReg is counter-productive, and that in the context of SSL, learning the projector is rather useful for satisfying the invariance criterion, which is absent in the ICA experiments. VCReg is rather optimized by the encoder $f$.

## 6    Conclusion: what is the interest of pairwise independence?

This work claims that SSL projectors enforce pairwise independence. But, is it a desirable property in a representation? For example, Wang & Jordan (2021) claim that mutual independence can be detrimental to learn disentangled representations. Pairwise independence may nevertheless be beneficial for representations to be linearly probed: Table 3 suggests that given two representations with similar test accuracy on ImageNet, the one with significantly lower HSIC performs better on some downstream tasks on linear evaluation, a common requirement for SSL representations. Showing the projector to be responsible for it, and demonstrating HSIC to be useful for model selection would be important results we leave for future work. Finally, this work focuses on random weights yet opens the way to weight distributions that are closer to training parameters in practice which, for example, have been characterized for overparameterized networks (Jacot et al., 2018).

Table 3: Test accuracy of SSL representations on common downstream tasks (linear evaluation).

| | HSIC | ImageNet | Places205 | iNat | CO3D | CO3D video |
|---|---|---|---|---|---|---|
| VICReg | 0.337 | 68.0 | 46.5 ± 0.2 | 27.6 ± 0.1 | 97.7 ± 0.1 | 98.1 ± 0.1 |
| VICReg, no BN in proj. | 0.239 | 67.6 | 47.1 ± 0.1 | 29.9 ± 0.2 | 98.0 ± 0.1 | 98.53 ± 0.03 |

## 7 REPRODUCIBILITY STATEMENT

For theoretical results, explanations on assumptions are provided in Section 3 and proofs are displayed either in the main paper or in Appendix B.1. The results are verified experimentally in Section 5.

Implementations of the algorithms used (HSIC, dHSIC, Randomized Batch Normalization, Linear ICA, and PNL ICA) are provided in Appendix D, and our code will be released if the paper is accepted.

Our experimental setup is detailed either in the main paper or in Appendix E. The latter include architectural details, augmentations, optimizer with learning rate and batch size, and grid search for hyper-parameters. The numbers used to produce the Figures in 5 are provided in C.1.

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

# A    Further Background

## A.1    VCReg methods

**Learning visual representations**    has a long history in machine learning. Shortly after the advent of convolutional neural networks (CNN), it was common to train a model on a supervised task such as ImageNet before removing the classifier and use the remaining model to produce features for downstream tasks (e.g., classification, segmentation), a technique coined transfer learning. Then, strategies to learn representations without labelled dataset by enforcing invariant embeddings emerged (Misra & Maaten, 2020; Chen et al., 2020). Joint embedding methods can be divided in two categories. Contrastive learning (e.g., SimCLR (Chen et al., 2020)) pulls together representations of two augmented versions of the same image while pushing away the representation of this image from the representation of different images. Non-contrastive learning (e.g., DINO (Caron et al., 2021)) pulls together representations of two augmented versions of the same image while avoiding collapse using different techniques. The frameworks considered in this work belong to the latter category. Non-contrastive self-supervised representation learning of images has a few peculiarities:

- Data augmentations are central, at least when it comes to learn from ImageNet, and must be carefully chosen.
- Most methods can be used either with Vision Transformers (Dosovitskiy et al., 2020), or with CNNs such as ResNets (He et al., 2016). In this work, we focus on ResNets.
- Model selection is usually performed via the Top1 accuracy on the validation set of ImageNet using a linear classifier on top of the learned representation. Outside of joint embedding, methods such as Masked-Auto-Encoder (He et al., 2021) perform poorly in linear evaluation while delivering excellent performance when fine-tuned on the downstream task.

**The crucial role of the projector.**    It was found by Chen et al. (2020) that adding a MLP on top of the encoder (removed after the training) significantly improved the quality of the learned representation. For example, in 100 epochs, VICReg without projector would achieve $48\%$ validation accuracy on ImageNet instead of 68 and SimCLR $50\%$ instead of $68\%$. Since then, a few theoretical work attempted to explain SSL with joint embeddings. Jing et al. (2021) study the role of linear projectors with restricted augmentations, while other works such as (Huang et al., 2021; Wang & Isola, 2020; HaoChen et al., 2021; Tian et al., 2020; Wang & Liu, 2021) only consider an encoder without projector in their theoretical analysis. Finally, Cosentino et al. (2022) consider a MLP projector in a very restricted context where data augmentations are Lie group transformations.

**BarlowTwins.**    Zbontar et al. (2021) propose a slightly different approach based on regularizing $\boldsymbol{C}$, the $P \times P$ cross-correlation matrix between $\boldsymbol{Z}_{\text{left}}$ and $\boldsymbol{Z}_{\text{right}}$ by optimizing

$$\mathcal{L}_{\text{BT}} = \sum_{k=1}^{K}((\boldsymbol{C})_{k,k} - 1)^2 + \alpha \sum_{k' \neq k}(\boldsymbol{C})_{k,k'}^2 . \tag{7}$$

Here, the leftmost term corresponds to regularizing the cross-correlation of the same feature in the two views to be unit, while the rightmost term regularizes the cross-correlation between pairs of different features in the two views. Importantly, $(\boldsymbol{C})_{i,j}$ falls back to measuring the cosine similarity between the $i^{\text{th}}$ column of $\boldsymbol{Z}_{\text{left}}$ and the $j^{\text{th}}$ column of $\boldsymbol{Z}_{\text{right}}$ i.e. $(\boldsymbol{C})_{i,j} = \frac{\langle(\boldsymbol{Z}_{\text{left}})_{.,i},(\boldsymbol{Z}_{\text{right}})_{.,j}\rangle}{\|(\boldsymbol{Z}_{\text{left}})_{.,i}\|_2\|(\boldsymbol{Z}_{\text{right}})_{.,j}\|_2}$. Hence, the leftmost term is also an invariance term: all features must be similar for both views.

**W-MSE.**    Ermolov et al. (2021) use the following loss

$$\min 2 - 2\frac{\langle\boldsymbol{Z}_{\text{left}}, \boldsymbol{Z}_{\text{right}}\rangle}{\|\boldsymbol{Z}_{\text{left}}\|_2\|\boldsymbol{Z}_{\text{right}}\|_2}$$
$$\text{s.t. } \text{Cov}(\boldsymbol{Z}_{\text{left}}) = Id, \text{Cov}(\boldsymbol{Z}_{\text{right}}) = Id.$$

where the cosine similarity can be replaced by the Euclidean distance.

## A.2 Independent Component Analysis

The goal of ICA (Comon, 1994) is to find a transformation of a random vector $\boldsymbol{Y}$[3] which minimizes the statistical dependence of its components. Its simplest instance is linear ICA, and is motivated by problems such as the cocktail-party, where $\boldsymbol{Y} \in \mathbb{R}^{N \times D}$ typically results from a linear transformation of independent sources $\boldsymbol{S}$ (*e.g* overlapping voices) one wants to recover. Formally, $\boldsymbol{Y} = \boldsymbol{SA}$, with $\boldsymbol{A} \in \mathbb{R}^{D \times D}$ an unknown mixing matrix that can therefore not be inverted. Instead, the linear ICA approach searches for $\boldsymbol{M} \in \mathbb{R}^{D \times D}$ such that $\boldsymbol{YM} = \boldsymbol{S}$ by maximizing the statistical *mutual* independence between the components of $\boldsymbol{YM}$. Being able to recover $\boldsymbol{S}$ is a property known as identifiability.

## B Proofs

### B.1 Proof of lemma 2: Random Linear Projectors maximize pairwise independence

*Proof.* Let's consider the linear regime for which $g((\boldsymbol{X})_{:,i}) = (\boldsymbol{X})_{:,i} \boldsymbol{w}_i^T$ with orthogonal weights i.e. $\langle \boldsymbol{w}_i, \boldsymbol{w}_j \rangle = 1_{\{i=j\}}$. This assumption is realistic since we assume that $\boldsymbol{w} \in \mathbb{R}^K$ with $K$ very large, and that those are randomly sampled. From that, we then have

$$\sum_{i \neq j} \text{HSIC}((\boldsymbol{X})_{:,i}, (\boldsymbol{X})_{:,j}) = \frac{1}{(N-1)^2} \sum_{i \neq j} \text{Tr}(g((\boldsymbol{X})_{:,i})g((\boldsymbol{X})_{:,i})^T \boldsymbol{H} g((\boldsymbol{X})_{:,j})g((\boldsymbol{X})_{:,j})^T \boldsymbol{H})$$

$$= \frac{1}{(N-1)^2} \sum_{i \neq j} \|g((\boldsymbol{X})_{:,i})^T \boldsymbol{H} g((\boldsymbol{X})_{:,j})\|_F^2,$$

we will now push the sum inside the norm by considering the following equality:

$$\|\sum_{i \neq j} f(i,j)\|_F^2 = \sum_{i \neq j} \|f(i,j)\|_F^2 + \sum_{i \neq j} \sum_{k \neq \ell, (i,j) \neq (k,\ell)} \text{Tr}\left(f(i,j)^T f(k,\ell)\right) = \sum_{i \neq j} \|f(i,j)\|_F^2,$$

since in our case

$$f(i,j) = g((\boldsymbol{X})_{:,i})^T \boldsymbol{H} g((\boldsymbol{X})_{:,j}) = \boldsymbol{w}_i (\boldsymbol{X})_{:,i}^T \boldsymbol{H} (\boldsymbol{X})_{:,j} \boldsymbol{w}_j^T,$$

$$f(i,j)^T f(k,\ell) = \boldsymbol{w}_j (\boldsymbol{X})_{:,j}^T \boldsymbol{H} (\boldsymbol{X})_{:,i} \boldsymbol{w}_i^T \boldsymbol{w}_k (\boldsymbol{X})_{:,k}^T \boldsymbol{H} (\boldsymbol{X})_{:,\ell} \boldsymbol{w}_\ell^T,$$

$$\text{Tr}(f(i,j)^T f(k,\ell)) = 1_{\{i=k \wedge j=\ell\}} (\boldsymbol{X})_{:,j}^T \boldsymbol{H} (\boldsymbol{X})_{:,i} (\boldsymbol{X})_{:,k}^T \boldsymbol{H} (\boldsymbol{X})_{:,\ell},$$

leading to

$$\sum_{i \neq j} \text{HSIC}((\boldsymbol{X})_{:,i}, (\boldsymbol{X})_{:,j}) = \frac{1}{(N-1)^2} \|\sum_{i \neq j} g((\boldsymbol{X})_{:,i})^T \boldsymbol{H} g((\boldsymbol{X})_{:,j})\|_F^2$$

$$= \frac{1}{(N-1)^2} \|(\sum_i g((\boldsymbol{X})_{:,i}))^T \boldsymbol{H} (\sum_j g((\boldsymbol{X})_{:,j})) - \sum_i g((\boldsymbol{X})_{:,i})^T \boldsymbol{H} g((\boldsymbol{X})_{:,i})\|_F^2$$

$$= \|\frac{1}{N-1} (\sum_i g((\boldsymbol{X})_{:,i}))^T \boldsymbol{H} (\sum_j g((\boldsymbol{X})_{:,j})) - \text{diag}(\text{Cov}(\boldsymbol{XW}))\|_F^2$$

$$= \|\text{Cov}(\boldsymbol{XW}) - \text{diag}(\text{Cov}(\boldsymbol{XW}))\|_F^2$$

$$= \sum_{i \neq j} \text{Cov}(\boldsymbol{XW})_{i,j}^2,$$

since we assume that all columns of $\boldsymbol{X}$ have same variance and that $\boldsymbol{W}$ is orthogonal we have for the pre-last equality

$$\frac{1}{N-1} \sum_i g((\boldsymbol{X})_{:,i})^T \boldsymbol{H} g((\boldsymbol{X})_{:,i}) = \text{Var}((\boldsymbol{X})_{:,1}) \sum_i \boldsymbol{w}_i \boldsymbol{w}_i^T = \text{Var}((\boldsymbol{X})_{:,1}),$$

and for the last equality we use the fact that since $g$ is linear, $\sum_i g((\boldsymbol{X})_{:,i}) = \sum_i (\boldsymbol{X})_{:,i} \boldsymbol{w}_i^T = \boldsymbol{XW}$ with $\boldsymbol{W} = [\boldsymbol{w}_1, \ldots, \boldsymbol{w}_K]^T$.

$\square$

---

[3] For simplicity, $\boldsymbol{Y}$ directly denotes $N$ empirical $D$-dimensional observations.

### B.2 ON THE EQUIVALENCE BETWEEN VICREG AND BARLOWTWINS OBJECTIVES

We can express Barlow Twins objective as:

$$\min \sum_{k=1}^{K} (\text{Cov}(\boldsymbol{Z}_{\text{left}}, \boldsymbol{Z}_{\text{right}})_{k,k} - 1)^2 + \alpha \sum_{k' \neq k} \text{Cov}(\boldsymbol{Z}_{\text{left}}, \boldsymbol{Z}_{\text{right}})_{k,k'}^2$$
$$\text{s.t. } \text{Cov}(\boldsymbol{Z}_{\text{left}}) = \boldsymbol{I}, \text{Cov}(\boldsymbol{Z}_{\text{right}}) = \boldsymbol{I}.$$

Assuming $\boldsymbol{Z}_{\text{left}}^T \boldsymbol{Z}_{\text{left}} = \boldsymbol{I}, \boldsymbol{Z}_{\text{right}}^T \boldsymbol{Z}_{\text{right}} = \boldsymbol{I}$ i.e. perfect minimization of the variance and covariance terms, we have

$$C_{i,j} = \frac{\langle (\boldsymbol{Z}_{\text{left}})_{:,i}, (\boldsymbol{Z}_{\text{left}})_{:,j} \rangle}{\|(\boldsymbol{Z}_{\text{left}})_{:,i}\|_2 \|(\boldsymbol{Z}_{\text{left}})_{:,j}\|_2} = \frac{1}{2} \langle (\boldsymbol{Z}_{\text{left}})_{:,i}, (\boldsymbol{Z}_{\text{right}})_{:,j} \rangle = -\|(\boldsymbol{Z}_{\text{left}})_{:,i} - (\boldsymbol{Z}_{\text{right}})_{:,j}\|_2^2 - 1,$$

and thus

$$\sum_i (C_{i,i} - 1)^2 = \sum_i \|(\boldsymbol{Z}_{\text{left}})_{:,i} - (\boldsymbol{Z}_{\text{right}})_{:,i}\|_2^4 = \|\boldsymbol{Z}_{\text{left}} - \boldsymbol{Z}_{\text{right}}\|_F^4 \propto I(\boldsymbol{Z}_{\text{left}}, \boldsymbol{Z}_{\text{right}}),$$

so we recover the invariance loss exactly with the diagonal terms of BarlowTwins. We now show this is actually enough to minimize this quantity to also minimize the off-diagonal terms:

$$\sum_{i \neq j} C_{i,j}^2 = \sum_{i \neq j} (1 - \frac{1}{2} \|(\boldsymbol{Z}_{\text{left}})_{:,i} - (\boldsymbol{Z}_{\text{right}})_{:,j}\|_2^2)^2$$

$$= \sum_{i \neq j} (1 - \frac{1}{2} \|(\boldsymbol{Z}_{\text{left}})_{:,i} - (\boldsymbol{Z}_{\text{left}})_{:,j} + (\boldsymbol{Z}_{\text{left}})_{:,j} - (\boldsymbol{Z}_{\text{right}})_{:,j}\|_2^2)^2$$

$$= \sum_{i \neq j} \Big( 1 - \frac{1}{2} \|(\boldsymbol{Z}_{\text{left}})_{:,i} - (\boldsymbol{Z}_{\text{left}})_{:,j}\|_2^2 - \frac{1}{2} \|(\boldsymbol{Z}_{\text{left}})_{:,j} - (\boldsymbol{Z}_{\text{right}})_{:,j}\|_2^2$$
$$- \langle (\boldsymbol{Z}_{\text{left}})_{:,i} - (\boldsymbol{Z}_{\text{left}})_{:,j}, (\boldsymbol{Z}_{\text{left}})_{:,j} - (\boldsymbol{Z}_{\text{right}})_{:,j} \rangle \Big)^2$$

$$= \sum_{i \neq j} \Big( - \frac{1}{2} \|(\boldsymbol{Z}_{\text{left}})_{:,j} - (\boldsymbol{Z}_{\text{right}})_{:,j}\|_2^2 - \langle (\boldsymbol{Z}_{\text{left}})_{:,i} - (\boldsymbol{Z}_{\text{left}})_{:,j}, (\boldsymbol{Z}_{\text{left}})_{:,j} - (\boldsymbol{Z}_{\text{right}})_{:,j} \rangle \Big)^2$$

$$= \sum_{i \neq j} \Big( \frac{1}{2} \|(\boldsymbol{Z}_{\text{left}})_{:,j} - (\boldsymbol{Z}_{\text{right}})_{:,j}\|_2^2 + \langle (\boldsymbol{Z}_{\text{left}})_{:,i} - (\boldsymbol{Z}_{\text{left}})_{:,j}, (\boldsymbol{Z}_{\text{left}})_{:,j} - (\boldsymbol{Z}_{\text{right}})_{:,j} \rangle \Big)^2$$

$$\leq \sum_{i \neq j} \Big( \frac{1}{2} \|(\boldsymbol{Z}_{\text{left}})_{:,j} - (\boldsymbol{Z}_{\text{right}})_{:,j}\|_2^2 + \|(\boldsymbol{Z}_{\text{left}})_{:,i} - (\boldsymbol{Z}_{\text{left}})_{:,j}\|_2^2 \|(\boldsymbol{Z}_{\text{left}})_{:,j} - (\boldsymbol{Z}_{\text{right}})_{:,j}\|_2^2 \Big)^2$$

$$= \frac{25(K-1)}{4} \|\boldsymbol{Z}_{\text{left}} - \boldsymbol{Z}_{\text{right}}\|_2^4,$$

hence minimizing the BarlowTwins loss with the explicit whitening constraint is equivalent to minimizing VICReg with explicit whitening constraint.

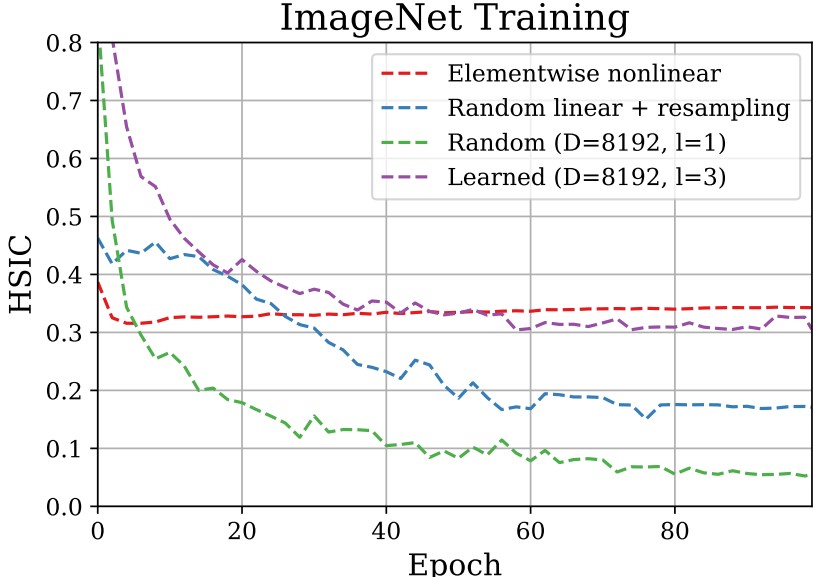

Figure 4: The projectors proposed in Section 3 better optimize HSIC than their classical counterpart.

## C    ADDITIONAL EXPERIMENTAL RESULTS

### C.1    HSIC OF REPRESENTATIONS LEARNED FROM IMAGENET

Table 4: Measure of pairwise dependence (the lower the better) and test accuracy for the projectors studied in Section 3. $L$ is the number of nonlinear projections in Lemma 1, $D$ is the width of the projector and $l$ is the number of layers.

| Projector type | Resampling | | No resampling | |
| --- | --- | --- | --- | --- |
| | Mean HSIC $\downarrow$ | ImageNet Top1 $\uparrow$ | Mean HSIC $\downarrow$ | ImageNet Top1 $\uparrow$ |
| No projector | N.A. | N.A. | 0.48 | 48.6 |
| Elementwise nonlinear ($L = 8$) | N.A. | N.A. | 0.34 | 51.0 |
| Random linear ($D = 16384$) | 0.16 | 54.3 | 0.13 | 54.2 |
| Random linear ($D = 8192$) | 0.14 | 53.6 | 0.30 | 52.9 |
| Random linear ($D = 4096$) | 0.16 | 52.3 | 0.24 | 52.9 |
| Random linear ($D = 2048$) | 0.18 | 51.7 | 0.38 | 52.3 |
| Random linear ($D = 1024$) | 0.27 | 51.1 | 0.39 | 52.0 |
| Random linear ($D = 512$) | 0.49 | 50.0 | 0.54 | 51.3 |
| Random ($D = 8192, l = 1$) | 0.06 | 50.9 | 0.48 | 53.6 |
| Random ($D = 8192, l = 2$) | 0.63 | 53.3 | 0.52 | 53.4 |
| Random ($D = 8192, l = 3$) | 0.56 | 52.8 | 0.62 | 54.9 |
| Learned ($D = 8192, l = 1$) | 0.17 | 58.4 | N.A. | N.A. |
| Learned ($D = 8192, l = 2$) | 0.36 | 65.1 | N.A. | N.A. |
| Learned ($D = 8192, l = 3$) | 0.30 | 68.4 | N.A. | N.A. |
| Same without BN | 0.22 | 68.1 | N.A. | N.A. |
| Same with Random BN | 0.31 | 69.1 | N.A. | N.A. |

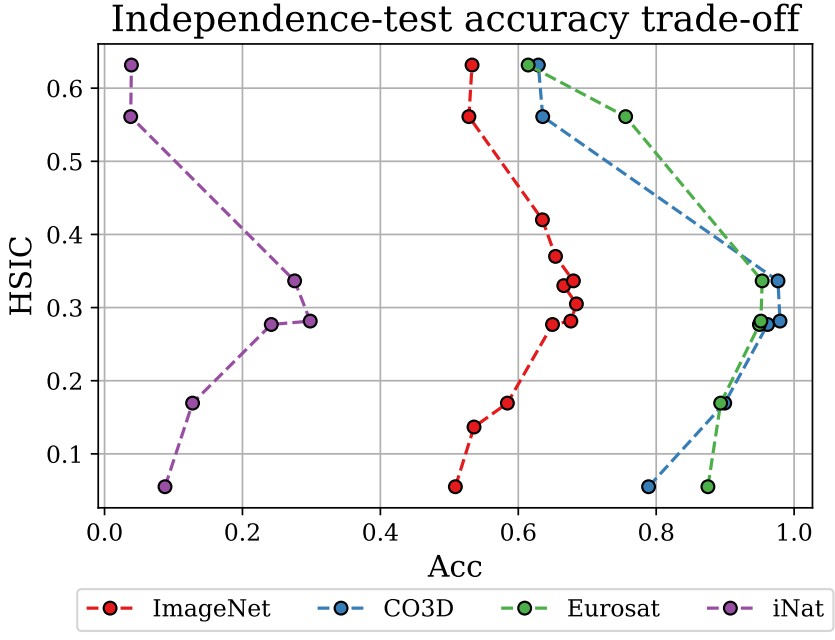

Figure 5: Minimizing HSIC is beneficial up to some extent, since HSIC does not guarantee that useful information is present in the representation. Beyond this limit, the performance decreases on ImageNet and on downstream tasks.

Table 5: VICReg with 2048-2048-2048 projector and varying covariance coefficient. Increasing this coefficient does not necessarily decreases HSIC. Indeed, the projector may start to optimize too much for the VC criterion thus getting trapped in bad solutions.

| Width | Mean HSIC ↓ | ImageNet Top1 ↑ |
|---|---|---|
| 2 | 0.33 | 66.6 |
| 4 | 0.42 | 66.7 |
| 6 | 0.37 | 66.6 |
| 8 | 0.29 | 66.3 |
| 10 | 0.28 | 66.5 |
| 12 | 0.37 | 66.2 |

Table 6: VICReg with learned MLP projector and varying sizes. The wider the projector, the lower HSIC, which conforms to our theoretical results in Section 3.

| Covariance coeff | Mean HSIC ↓ | ImageNet Top1 ↑ |
|---|---|---|
| 512 | 0.42 | 63.5 |
| 1024 | 0.37 | 65.4 |
| 2048 | 0.33 | 66.6 |
| 8192 | 0.31 | 68.4 |

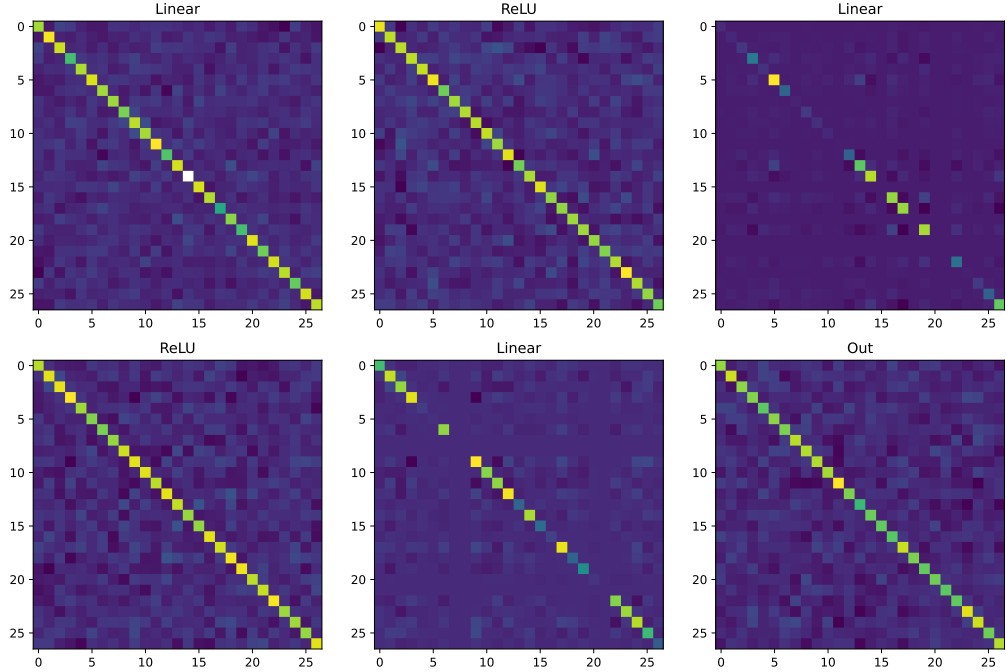

Figure 6: The covariance matrices for some features of each activation in a random VICReg projector (same as in Bardes et al. (2022) with width 8192) before corresponding hidden layer are close to diagonal, suggesting that each hidden layer output is implicitly VC-regularized.

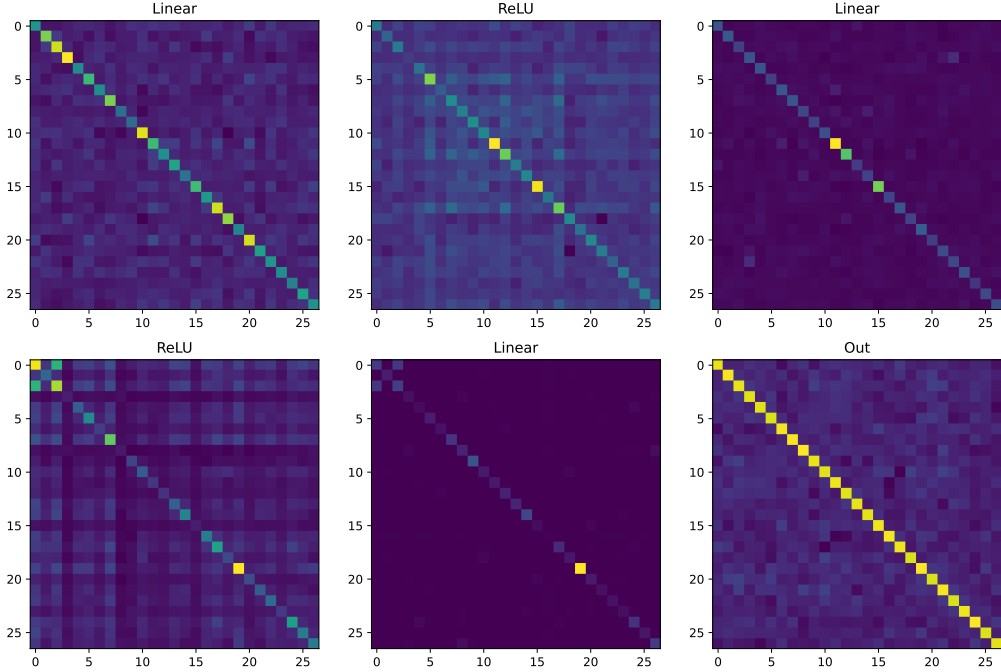

Figure 7: The covariance matrices for some features of each activation in a learned VICReg projector (same as in Bardes et al. (2022) with width 8192) before corresponding hidden layer are close to diagonal, suggesting that each hidden layer output is implicitly VC-regularized.

## D  IMPLEMENTATIONS

### D.1  HSIC EQ. (2)

```
1  def GaussianKernelMatrix(X, sigma):
2      pairwise_distances = torch.cdist(X, X)
3      return torch.exp( -pairwise_distances / (2 * sigma**2))
4
5  def HSIC(X_1, X_2, sigma_1, sigma_2):
6      N = x.size(0) # batch size, should be the same for X_1 and X_2
7      K_1 = GaussianKernelMatrix(X_1, sigma_1) # Gaussian kernel matrix of
         X_1 with bandwidth sigma_1
8      K_2 = GaussianKernelMatrix(X_2, sigma_2) # Gaussian kernel matrix of
         X_2 with bandwidth sigma_2
9      H = torch.eye(N) - 1.0 / N  # centering matrix
10     HSIC = torch.trace(K_1 @ H @ K_2 @ H)/((N-1)**2)
11     return HSIC
```

In our study, the bandwidth $\sigma$ of the Gaussian kernel is determined by the median of the distribution of pairwise euclidean distances between samples.

### D.2  DHSIC EQ. (4)

dHSIC can be estimated given empirical samples $X_1, \ldots, X_d$ with respective kernel matrices $K_1, \ldots, K_d$ as:

$$\mathrm{dHSIC}(X_1, \ldots, X_d) = \frac{1}{N^2} \sum_{i,j} (\bigodot_{k=1}^{d} K_k)_{i,j} + \frac{1}{N^{2d}} \prod_{k=1}^{d} \sum_{i,j} (K_k)_{i,j} - \frac{2}{N^{d+1}} \sum_{i} \bigodot_{k=1}^{d} \sum_{j} (K_k)_{i,j}.$$

When $d = 2$, the first term corresponds to a biased HSIC estimator. The implementation of dHSIC is given by:

```
1  import GaussianKernelMatrix # defined above
2
3  def dHSIC(X, sigma):
4      length = X.shape(0)
5      term_1 = 1.0
6      term_2 = 1.0
7      term_3 = 2.0 / length
8      for j in range(D):
9          K_j = GaussianKernelMatrix(X[j])
10         term_1 = torch.mul(term_1, K_j)
11         term_2 = 1.0 / length / length / term_2 * torch.sum(K_j)
12         term_3 = 1.0 / length * term_3 * K_j.sum(axis=0)
13
14     term_1 = (1.0 / length) ** 2 * torch.sum(term_1)
15     term_3 = torch.sum(term_3)
16
17     return term_1 + term_2 - term_3 # the three terms of the estimator
```

As these evaluations already take a few minutes, scaling it to significantly larger portions of components would be impractical.

### D.3  RANDOMIZED BATCH NORMALIZATION

```
1  class RandBatchNorm1d(torch.nn.BatchNorm1d):
2      def forward(input):
3          # the mean and variance term in classical batch norm are
4          # randomized
5          m = torch.randn_like(m) * self.estimator_mean.std +
6              self.estimator_mean.mean
7          std = torch.distributions.Gamma(
8              self.estimator_std.mean.square() / self.estimator_std.var,
9              self.estimator_std.mean / self.estimator_std.var,
10             ).sample()
```

```
11          return self.weight * (input - m) / torch.sqrt(std.square_() +
        self.eps) + self.bias
```

## D.4   LINEAR ICA MODEL 3

```
1  for p in projector.parameters(): # freeze the projector
2          p.requires_grad = False
3
4  for y in loader:
5      x = encoder(y) # matrix multiplication with M
6
7      z = projector(x) # compute embedding
8
9      C = torch.cov(z) # covariance matrix
10     loss = torch.MSE(C, torch.identity()) # VCReg loss
11     loss.backward()
12     optimizer.step()
13
14     projector.__init__()  # to resample the projector
15     for p in projector.parameters(): # freeze the projector
16          p.requires_grad = False
```

## D.5   NONLINEAR ICA MODEL 9

The implementation of our nonlinear ICA model 9 differs from 3 by the addition of a reconstruction constraint:

```
1  for p in projector.parameters(): # freeze the projector
2          p.requires_grad = False
3
4  for y in loader:
5      x = encoder(y)  # MLP encoder
6
7      z = projector(x) # compute embedding
8
9      y_rec = reconstructor(x) # MLP reconstructor
10
11     C = torch.cov(z) # covariance matrix
12     loss = torch.MSE(C, torch.identity()) + lmda * torch.MSE(y, y_rec) #
        VCReg loss with reconstruction
13     loss.backward()
14     optimizer.step()
15
16     projector.__init__()  # to resample the projector
17     for p in projector.parameters(): # freeze the projector
18          p.requires_grad = False
```

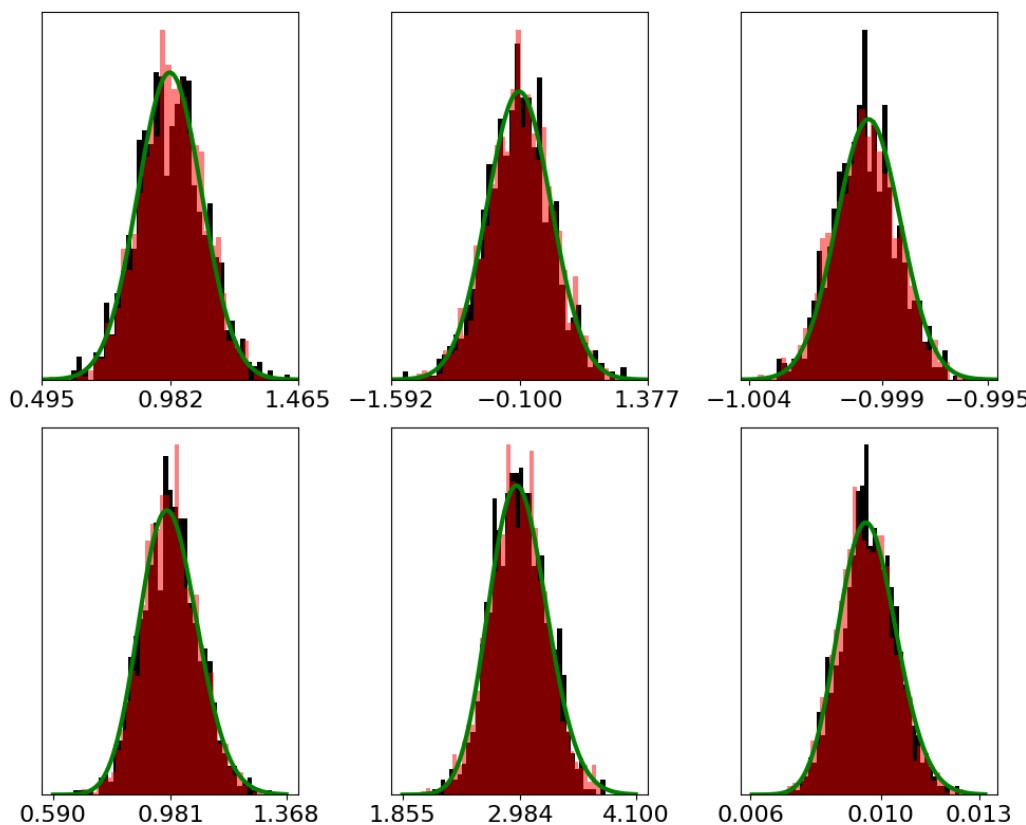

Figure 8: Distributions of $\mu^{\ell,k}$ (top row) and $\sigma^{\ell,k}$ (bottom row), which are the BN centering and scaling statistics computed at each mini-batch for three units in a DN layer with $\boldsymbol{W}^\ell \boldsymbol{z}^{\ell-1} \sim \mathcal{N}([1, 0, -1], \mathrm{diag}([1, 3, 0.1]))$ ($\ell$ is not relevant in this context). The empirical distributions in **black** are obtained by repeatedly sampling mini-batches of size $64$ from a training set of size $1000$. The analytical distributions in **green** can be obtained by analysis, and its empirical distributions in **red** closely match the training one.

## E  EXPERIMENTAL DETAILS

### E.1  DETAILS ON IMAGENET EXPERIMENTS

**Architectures and hyper-parameters.** We use a Resnet50 bottleneck, optimized during $100$ epochs with LARS and an initial learning rate of $0.3$ for all methods. The batch size is $1024$ for all methods except SimCLR. All methods have projector $8192 - 8192 - 8192$ except DINO and Supervised. Throughout the training, we evaluate the learned representation using an online linear classifier. Details specific to each method:

- Barlow Twins: for Experiment 5.1, the off-diagonal coefficient is $0.0051$.

- VICReg: for Experiment 5.1, the projector size is $8192 - 8192 - 8192$, Invariance coeff is $25$, Variance coeff is $25$ and is Covariance coeff $1$. For the rest of the experiments, and following Bardes et al. (2022), we only move the covariance coefficient when changing the size of the projector. We scale it in the square root of the output size of the projector in order to have similar magnitude for the covariance term for all projector sizes.

- SimCLR: for Experiment 5.1, we choose a higher batch size of $2048$ as SimCLR is sensitive to this parameter (Chen et al., 2020). The temperature is $0.15$.

- DINO: we use $8$ crops. The head has $4$ layers of widths $2048 - 2048 - 2048 - 256 - 65536$.

**Augmentations.** At train time, the resolution is 160 and we use

- `RandomHorizontalFlip()`.
- `ColorJitter(0.8, 0.4, 0.4, 0.2, 0.1)`.
- `Greyscale(0.2)`.
- `NormalizeImage` with ImageNet mean and standard deviation.
- `GaussianBlur()` with kernel size $(5, 9)$ and sigma $(0.1, 2)$.

At validation time, the resolution is 224 and the images are cropped and normalized with ImageNet mean and standard deviation.

### E.2 ICA SETUP

**Detailed setup.** In these experiments, we keep the same optimizer, train on 100 epochs and choose a batch size of 64 for both datasets. We tune the learning rate according to a logarithmic grid $[1.0, 10.0, 100.0]$ (then rescaled according to $\frac{lr \times batchsize}{256}$). The MLP can either be a SSL-like projector or simply a fully-connected layer followed by a ReLU. The wider the MLP, the better the result hence it does not require selection. For the synthetic dataset, the MLP has width 1024, and 8192 for the audio one.

- Linear ICA: We tune the standard and covariance coefficients according to a logarithmic grid $[1, 10, 100]$.
- PNL ICA: Our architecture for the nonlinear ICA experiments is presented in Figure 9. The encoder $f$ is a MLP with 3 layers and width 128. The decoder $h$ is a learnable nonlinearity (a MLP with 3 hidden layers and depth 16) followed by a fully-connected layer. We tune the standard, covariance and reconstruction coefficients according to a larger logarithmic grid $[1, 3, 10, 30, 100]$.

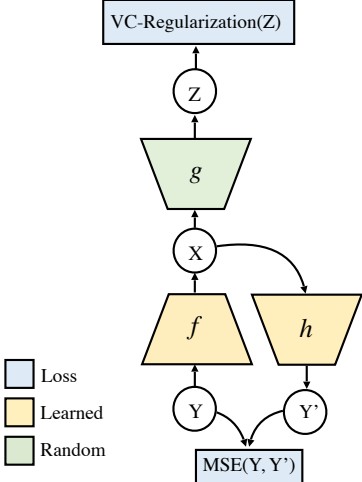

Figure 9: Nonlinear ICA model.

**Reconstructed sources for ICA.** In this paragraph, we provide ground truth sources for the synthetic data along with examples of reconstructed sources.

True sources

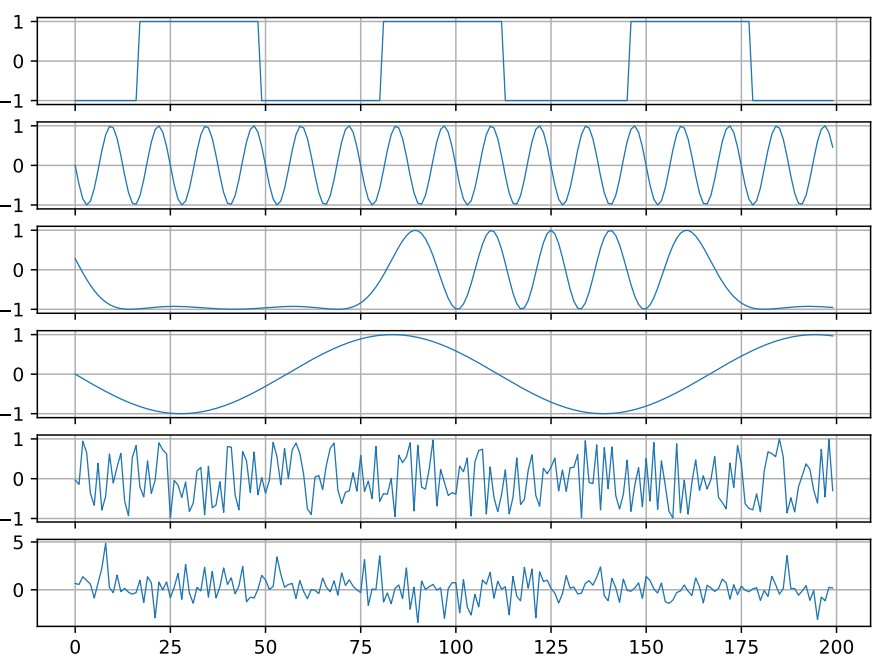

Figure 10: Data before mixing.

Mixed sources

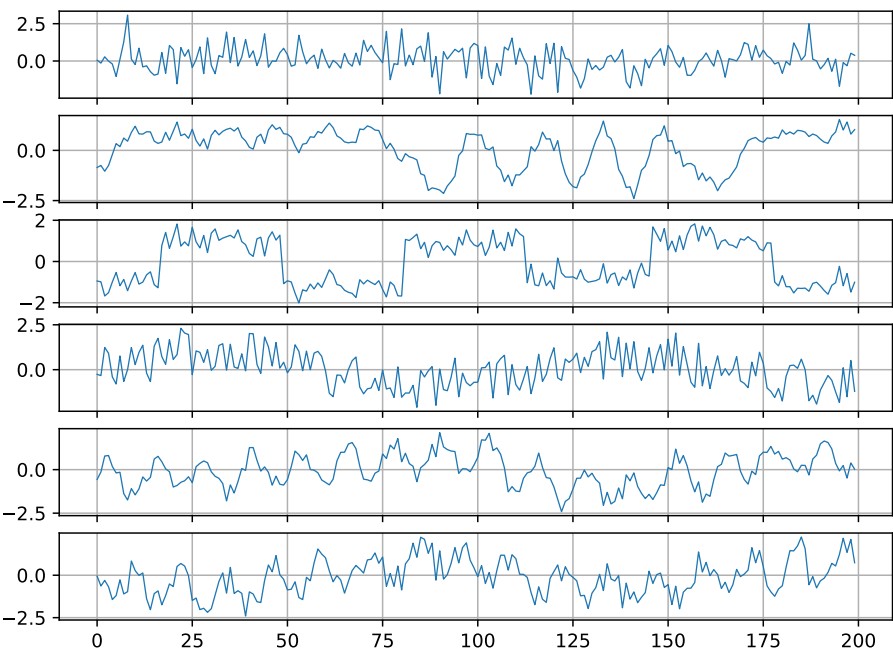

Figure 11: Data after mixing, before feeding to the various ICA models.

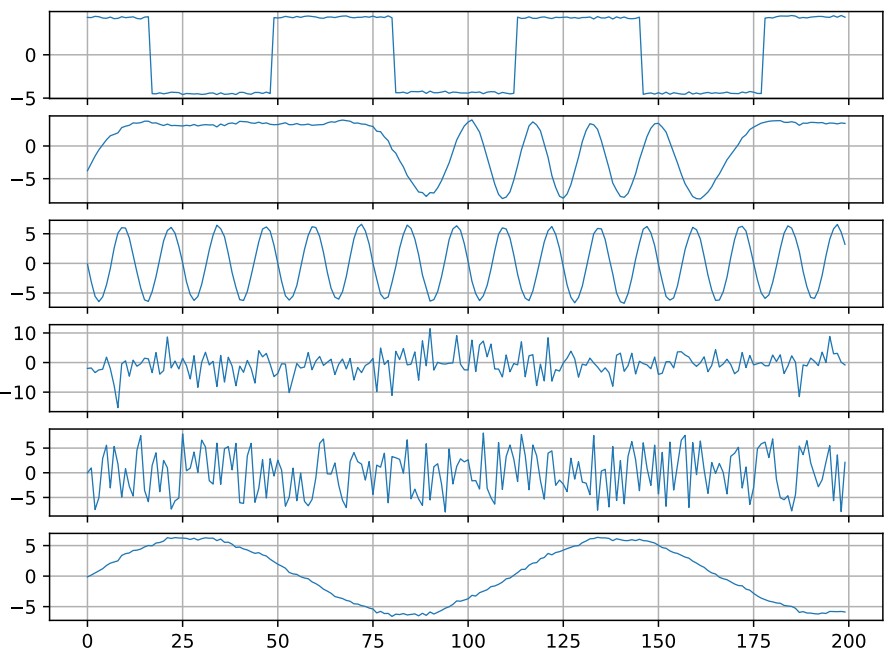

Figure 12: This level of max correlation is typically achieved by FastICA, VCReg or Anica.

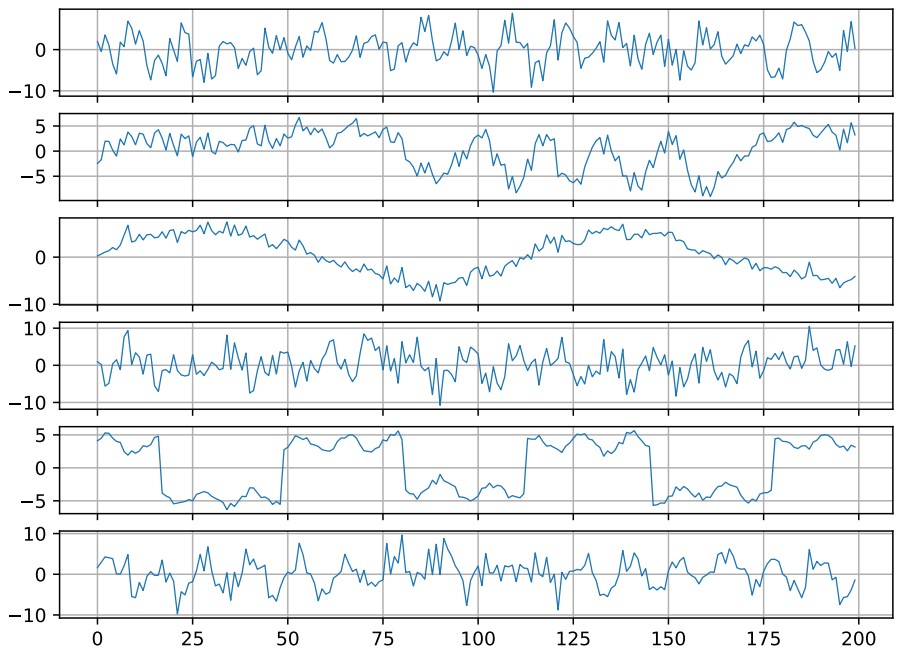

Figure 13: This level of max correlation is typically achieved by Whitening.

