# OpenReview forum: "Variance Covariance Regularization Enforces Pairwise Independence in Self-Supervised Representations"
_ICLR.cc/2023/Conference — Submitted to ICLR 2023_

### Official Review · Reviewer_EpBg · 2022-10-22

**Confidence:** 4
**Correctness:** 3
**Technical Novelty And Significance:** 2
**Empirical Novelty And Significance:** 2
**Recommendation:** 3

**Clarity, Quality, Novelty And Reproducibility:**

**Novelty**: it has been realized by previous papers that objectives such as Barlow Twins, W-MSE, VICReg all enforces pairwise independence. The findings in this paper is therefore unsurprising, though this paper is the first to formally characterize the precise relation between the training objective (i.e. the covariance term) and the independence measure (i.e. HSIC). However, I do not consider this as sufficient novelty.

**Reproducibility**: setup details are provided in Appendix E.

**Clarity**: The paper is clearly structured overall, though many writing details could be improved.
- below eq (4): what is "null dHSIC"?
- below eq (5): the first term in eq (5) does not "regularize the variance to be unit", but "to be at least unit".
- Lem 2: from the lemma statement itself, it's unclear what's the relationship between $g$ and $W$.
- Figure 2: please consider changing to colors with more contrast; for example, currently it's hard to distinguish "Learned MLP (l=3)" vs "Learned MLP (l=3) + no BN".
- Sec 5.3: $M, Y$ have dimension mismatch.
- Please consider some notation changes.
    - dHSIC: $X_D$ vs $\mathbf{X}_D$: one is a random var (a vector), the other is a sample matrix?
    - $K$ is used to denote both a kernel (e.g. in Lem 1) and a dimension (e.g. in Lem 2's proof).
- Please proofread for typos and grammar mistakes.



**Strength And Weaknesses:**

The paper is centered around pairwise independence of the features. I find the paper clearly structured overall, and there is sufficient discussion on related works.

However, I'm unconvinced why this is a property of interests or desirable. My current impression is that fully collapsing is certainly not good, but neither is full independence:
- Empirical findings show that more independence does not always mean better accuracy.
       - For example, random projectors usually get lower HSIC than learned projectors, but learned projectors get better accuracy.
- dHSIC plateaus during training, which suggests that there may be a mismatch between the SSL objective and pairwise independence.

That said, pairwise independence may be useful for model selection, if there is a clear correlation between HSIC and test accuracy.
- The empirical evidence is Figure 2 (top right), which I currently find questionable: how many replicates/runs are in each dot of the plot? If there's only a single run, then the trends may be due to variances in random seeds.

A clarification question about Lem 1: There seems to be a discrepancy of taking $L=1$ in the proof, and taking $L$ to be large when approximating the random kernel. How does the proof handle $L \neq 1$? I also find the notations confusing: what does $1-I_D$ mean (i.e. subtracting a matrix from a number)? Is $\otimes$ here denoting the tensor product?


**Summary Of The Paper:**

This paper studies a proposed VCReg loss and its relation to pairwise independence of features.

Theoretically, the paper equates the covariance term in VCReg loss to HSIC, for both arbitrary nonlinear point-wise function and sufficiently wide random linear transformation.

Empirically, the paper studies how properties of projection head affect pairwise independence. The results suggest that shallower and wider projectors should be used, and that a random projector suffices if we are only concerned about pairwise independence. Moreover, since VCReg enforces pairwise independence which is sufficient for linear ICA, the paper finds that VCReg is competitive with specialized algorithms such as Fast ICA.

A major concern though is that it is unclear why pairwise independence is an interesting property to study, since the paper finds that it does not correlates well with downstream performance.

**Summary Of The Review:**

This paper is centered around pairwise independence, however, currently it's unclear to me why this is an interesting property to study for representation learning, especially since the empirical results already show that there is no clear correlation between HSIC and downstream accuracy.

The paper is clearly structured overall, but many details could be improved for better readability.

---

> ### Author Response · Authors · 2022-11-16
> **Interest of pairwise independence**
>
> We thank the reviewer for the feedback. In our general answer, we explain why pairwise independence (HSIC) could in fact be a favorable property in a representation. We revised the experiments section which discusses the relationship between HSIC and test accuracy, as well as our conclusion which provides evidence of HSIC benefits.
>
> ### It has been realized by previous papers that objectives such as Barlow Twins, W-MSE, VICReg all enforces pairwise independence.
>
> As far as we are aware, existing studies have demonstrated that those methods provide some degree of decorelation/independence in the projector’s output (due to their loss function). In contrast, we are studying the input of the projector (which is the data representation of interest since the projector network is removed after SSL training) and in particular we obtain that in the projector’s input space, pairwise independent is enforced from VCReg being applied in the projector’s output space. We thus believe that those findings are complementary to the mentioned ones.
>
> ### dHSIC plateaus during training
>
> This is one of the claims of our work: SSL projectors combined to VC minimize pairwise independence (HSIC) but not mutual independence (dHSIC) of the components in the representation. As a result, dHSIC is not expected to strictly decrease (as opposed to HSIC). We provided computations of dHSIC for completeness and to pinpoint to the fact that (as pointed out by the reviewer) projectors in SSL do not strictly enforce dHSIC (we have made sure to clarify this distinction in the revised manuscript).
>
> ### Number of runs
>
> We are committed to provide runs with standard deviations if the paper is accepted; we did not manage to provide those in addition to all the new experiments we performed trying to answer to the main comments shared across all reviewers. We believe however that our experiments already provide a large amount of experiments (across projector configurations and hyper-parameters) that demonstrate how our findings hold consistently and that no cherry-picking was done.
>
> ### Clarification on Lemma 1
>
> The proof holds for $L = 1$ and above. When $L > 1$, $Z$ is a $N \times DL$ matrix and $g(X)_{:,i}$ is a $N \times L$ matrix.
>
> When $L = 1$, $Z$ is a $N \times D$ matrix and  $g(X)_{:,i}$  a $N$ vector.
>
> In both cases, $Cov(g(X)_{:,i})$  is a $N \times N$ matrix.
>
> $1$ is the matrix of ones while $I_D$ is the identity matrix.

---

> > ### Comment · Reviewer_EpBg · 2022-11-17
> > **Thank you for your response**
> >
> > I thank the authors for their responses and clarifications.
> > I understand that one contribution of the paper is to understand what the projector does, but showing that it encourages pairwise independence without well justifying why such property is interesting is not satisfying.
> >
> > My main concern remains that the motivation of studying pairwise independence is unclear; for example, there is no clear relation between HSIC and downstream performance. While I appreciate the additional table in the general reply / last paragraph of the revised submission, it has only 1 pair of comparisons and is hence not convincing enough.

---

> > > ### Author Response · Authors · 2022-11-19
> > > **Relationship between HSIC and downstream performance and more pair comparisons**
> > >
> > > We thank the reviewer for their response. We would like to point out the reviewer Figure 5 in Appendix C we had added to our revision, which demonstrates that in all cases, HSIC does correlate well with downstream performance when its value is large. Then, as HSIC gets very small, degenerate solutions (e.g. the encoder overfits for minimizing VC hence produces meaningless embeddings) start to appear, breaking the well defined relation that was present. Overall, for models with similar test accuracy, low HSIC is preferred (Table 3 in conclusion) (taking test accuracy into account allows to discard degenerate solutions before model selection).
> > >
> > > That being said, we would like to remind the reviewer that their critic on the motivation for pairwise independence seems excessive to us. As stated repeatedly in our manuscript and in our rebuttal:
> > > - In SSL, pairwise independence is a beneficial property if one is interested in disentanglement ([1] demonstrates that factors in real-world data tend to be pairwise independent), or in providing principles to improve our understanding of the role of the projector in SSL.
> > > - Besides, our first experiments suggest that for similar models (same architecture and similar test accuracy), lower HSIC entails better linear probing on downstream tasks, opening the way to a more practical motivation of our work. We believe a detailed analysis of this phenomenon to be beyond the scope of the present work. Regarding Table 3, we provide one more comparison pair for an architecture with a projector width of 2048. We are committed to continue providing more pair comparisons in the camera ready if the paper is accepted.
> > > -  Our finding is eye opening beyond deep learning (see ICA experiments).
> > >
> > > |                            | HSIC  | ImageNet | Places205 | iNat      | CO3D      | CO3D video |
> > > |----------------------------|-------|----------|-----------|-----------|-----------|------------|
> > > | VICReg (width=2048, covariance coefficient=2) | 0.313 | **66.6**     | 45.3+-0.2 | **25.7+-0.2** | 96.8+-0.1 | 98.0+-0.03 |
> > > | VICreg (width=2048, covariance coefficient=8) | 0.300 | 66.4     | **45.7+-0.2** | 25.1+-0.1 | 96.8+-0.1 | **98.1+-0.04** |
> > >
> > > [1] Li et al., Learning Disentangled Representation with Pairwise Independence (2019)

---

### Official Review · Reviewer_NhDr · 2022-10-23

**Confidence:** 5
**Correctness:** 3
**Technical Novelty And Significance:** 3
**Empirical Novelty And Significance:** 3
**Recommendation:** 6

**Clarity, Quality, Novelty And Reproducibility:**


The quality and clarity are median. The originality is good.


**Details Of Ethics Concerns:**


There is no ethics concern.


**Strength And Weaknesses:**



(Positive) The success or failure of self-supervised learning is still a mystery. This article attempts to provide some explanation for the regularization involved, which is highly encouraged.


(Positive) The paper demonstrates that regularizing the projector's output helps to encourage independence between the projector's input features. This is very interesting.


(Positive) The article also analyzes that the depth and width of the projector also affect the independence of its input features. This is also included as part of the theoretical framework presented in this paper.

(Positive) The method in this paper is very simple and clean.

(Negative) The theory in this paper (including the central limit law) relies on many random projections. But this does not fit perfectly with both self-supervised learning and supervised learning, whose parameters are to be learned. This creates a huge gap between its theoretical and practical needs.


(Negative) The most worrying thing is that low HSIC does not bring high test accuracy. For example, it is demonstrated in the paper that both random projection and resampling reduce HSIC (Figure 2). But what's the use of this? We cannot improve test accuracy with these strategies. Although the authors say, "the correlation is true within the same method, see Figure 2 (top right)", this is misleading. The x-axis is obtained by increasing the width. It is already a consensus that the test accuracy can be improved by increasing the width, which was not first discovered by this article. So, we did not learn much useful knowledge from this paper to help self-supervised learning.

(Negative) The analysis in this paper does not yield very useful insights into the improvement of self-supervised learning or supervised learning. It can neither fundamentally explain why existing self-supervised learning works nor provide effective guidance on how to improve the efficiency of self-supervised learning in the future.

(Negative) From an HSIC perspective, the results in Figure 1 are good. But this is only for HSIC.

(Negative) Learning the projector is not necessary to obtain pairwise independence and vice versa.

(Positive) I am happy to see that modifying Batch Normalization to get closer to a universal kernel in lemma one yields significant test accuracy gains for VICReg over Bardes et al. (2022).


(Negative)  The authors said that projector capacity should rather be increased via width than via depth: adding layers can be detrimental to HSIC as seen above but also to test accuracy. Where is the evidence of "detrimental to the test accuracy"?


(Positive) I am happy to see that "Learning the projector does not enforce pairwise independence."


**Summary Of The Paper:**


Summary:
This paper provides some explanations for regularization in self-supervised learning algorithms and, to some extent, other phenomena beyond self-supervised learning. Its analysis is reasonable and comprehensive. However, this analysis does not yield very useful insights into the improvement of self-supervised learning or supervised learning. It can neither fundamentally explain why existing self-supervised learning works nor provide effective guidance on how to improve the efficiency of self-supervised learning in the future. Overall, this is an article on the borderline, leaning toward being accepted.


**Summary Of The Review:**


See "Summary Of The Paper." This article is an above-average article and deserves a weak acceptance.

---

> ### Author Response · Authors · 2022-11-16
> **Motivation of the work with respect to SSL**
>
> We thank the reviewer for the thorough feedback. In our general answer, we explain why assuming random projectors is not such a big assumption, why our work could be an important step towards understanding the projector in SSL with joint embeddings, and why pairwise independence (HSIC) could in fact be a favorable property in a representation. We made sure to answer these concerns in our revised introduction and conclusion, and provided new experiments.
>
> ### Figure 1 only shows HSIC decrease.
>
> Does the reviewer refer to the fact that dHSIC plateaus? If so, this aligns with our claim that the projector combined with VC optimizes pairwise independence (HSIC) but not mutual independence (dHSIC).
>
> ### Learning the projector is not equivalent to obtaining pairwise independence
>
> We are not sure why this is a negative side of our work (in fact, this is rather one of the conclusions): could the reviewer elaborate on this comment? If the reviewer refers to the fact that in actual SSL the projector is learned and thus that our theoretical findings do not exactly match this scenario, we should point out that we have made sure to clearly outline the limitations of our work (restricted to random projector) and that despite this setting, our work is the first to provide a significant step towards a more principled understanding of the role of MLP projectors in SSL (see Appendix A for a thorough review of existing studies). We agree with the reviewer that there is room for further improvements and to precisely quantify the impact of training (as we pointed out in the revised conclusion).
>
> ### No evidence that adding layers to the projector is detrimental to test accuracy.
>
> This observation comes from Appalaraju et al., which evaluate MoCo with 2 layers and with 4 layers and observe decreased performance while [3] observe saturation or decrease with 4 layers for SimCLR (Table 2). We added the second reference in the manuscript.
>
> [3] Chen et al., Intriguing Properties of Contrastive Losses (2021)

---

> ### Comment · Reviewer_NhDr · 2022-12-11
> **Responses to the authors' responses:**
>
>
> I am very grateful to the authors for taking my concerns seriously. Although I still feel that the authors did not fully convince me of my two biggest concerns (the authors summarized them in the general answer), the authors did partially address my concerns. I am delighted that the authors show that a lower HSIC is helpful for downstream tasks. This one really hit me. Regarding random projectors, if the authors can give a theoretical explanation, I will be more convinced.
>
> I also read the excellent and valuable comments and suggestions from other reviewers. I obviously feel that authors should listen to reviewers, especially accepting negative comments and improving the paper.
>
> To sum up, this article is an above-average article with some advantages, but there are also limitations. I will keep my rating unchanged and recommend acceptance of this paper.

---

> > ### Author Response · Authors · 2022-12-12
> > **Thank you for your feedback**
> >
> > We deeply thank the reviewer for taking time to assess our rebuttal. Regarding random projectors, the assumption of random weights (even during training) is not unusual in the litterature: we already mention the Random Features paper by Rahimi and Recht (2007), [1] models weight evolution during training as a random walk and, as [2], assumes independent weights with gaussian distribution (the distribution evolves during the training). We believe this makes even more sense in the present work since our experiments on ICA show that optimizing the projector for VC leads to a collapse: the projector rather try to optimize the invariance term (we propose an explicit trivial solution in our general comment).
> >
> > [1] Franchi et al., TRADI: Tracking deep neural network weight
> > distributions for uncertainty estimation (2019)
> >
> > [2] Blundell et al. Weight Uncertainty in Neural Networks (2015)

---

### Official Review · Reviewer_MKGJ · 2022-10-24

**Confidence:** 4
**Correctness:** 3
**Technical Novelty And Significance:** 2
**Empirical Novelty And Significance:** Not applicable
**Recommendation:** 3

**Clarity, Quality, Novelty And Reproducibility:**

I believe the work is likely technically correct. Modulo some trouble parsing notation, it is clear what the main result is and why it's true.  As far as I can tell, this submission does have supplementary material available, so I can't comment on reproducibility.

With respect to clarity, I think the paper could use some polish. Illustrative examples:

1. The word "variable" is used throughout to mean "element of the representation X", but as far as I can tell this isn't made explicit. I was confused for quite a while about what exactly independence was being enforced between.

2. Section 2.2 and section 3 have conflicting definitions for *Z*. The definition of VCREG in section 2.2 actually only makes sense with the section 3 definition of Z.

3. The statement of Theorem 3 omits some apparently key conditions---e.g., presumably the result is only true in the limit of infinite layer width.

**Strength And Weaknesses:**

Strengths: the basic observation is fairly clean and, I believe, correct (though I have not checked the proofs).

Weaknesses:

The main weakness I see is a lack of motivation. It seems pretty clear a priori that the regularization term is pushing towards a relaxed notion of independence between elements of X. The main contribution of the paper is then an argument that, loosely, this relaxation is not too severe and we can view the regularizer as targeting independence between elements of the representation.

What's not clear to me is: why is this independence desirable? What might go wrong with a weaker notion of independence? Does this somehow help us understand when VCREG should or should not be preferred to other techniques for preventing embedding collapse? Without some answer to these questions, it's not clear to me why the technical results here are insightful.

These questions seem particularly salient given that 1. the experiments show that the learned representations don't actually have elementwise independence (e.g., in figure 1 measured HSIC never gets close to 0), and 2. the existence of arguments that elementwise independence of representations is anyways undesirable---see e.g., https://arxiv.org/abs/2109.03795

Closely related: I was generally unclear about what questions the experiments were meant to answer.

More minor: the results only hold for a variant of the encoder learning where the function g taking X to Z is random (i.e., not learned). This is clear in the relevant section, but not appropriately flagged in the introduction. Indeed, the authors present empirical results showing that training this map does indeed cause the independence to fail.

I also have some relatively minor problems with the writing, which is not yet fully baked.

**Summary Of The Paper:**

This paper studies VCREG, a particular regularization function used for contrastive representation learning. The purpose of this term is to prevent embedding collapse, where all inputs map to the same embedding (a trivial way to satisfy that embeddings corresponding to different views of the same object should map to the same embedding). In this paper, they model the embedding procedure as operating in two stages: first, an encoder produces a representation X for each input, then X is passed to a neural network g to produce a higher dimensional embedding Z=g(X). The VCREG term aims to enforce 0 covariance between the elements of Z. The main observation of the paper is that 0 covariance between elements of Z can be viewed (approximately) as enforcing independence between the elements of X.

**Summary Of The Review:**

This paper is not yet ready for publication.

---

> ### Author Response · Authors · 2022-11-16
> **Motivation of this work and interest of independence**
>
> We thank the reviewer for the thorough feedback. In our general comment, we explain why our work could be an important step towards understanding the projector in SSL with joint embeddings, and why pairwise independence could in fact be a favorable property in a representation. We added these points in the revised introduction and conclusion. We emphasize that precisely characterizing the notion of independence optimized (pairwise independence vs. mutual independence, for which kernel and behavior w.r.t. the hyperparameters) was not trivial. We improved the clarity of the manuscript following the reviewer’s suggestion.
>
> ### Learned representations do not have pairwise independence.
>
> Since HSIC is a regularization term, VICReg reaches an equilibrium between VC and the invariance term, so it can be expected that it does not reach 0. In Figure 1, HSIC is aggregated for all pairs of variables, hence can be low for some pairs and high for others. According to our independence tests, a significant fraction of pairs is independent, see Table 1. Also, note that the learning rate is scheduled and tends to 0, hence it is not surprising for HSIC plateaus near the end of the training. When training on more than 100 epochs, lower HSIC are reached.
>
> ### There are arguments against elementwise independence (https://arxiv.org/pdf/2109.03795.pdf).
>
> In their work, Wang and Jordan claim that methods for disentangled representation learning such as beta-TCVAE and FactorVAE may not be appropriate since they seek factorial distributions for factors of variation, i.e., mutually independent components for the representation. However, our work demonstrates that the projector combined with VC optimizes pairwise independence of the components in the representation (the resulting distribution p(x_1, …, x_d) is not factorial, only p(x_i, x_j) is). We added this relevant reference with our comment in the revised manuscript.
>
> ### Which questions do we want to answer with our experiments?
>
> Generally, our experiments seek to characterize the notion of independence enforced by an SSL projector combined with VCReg (pairwise vs. mutual, strength, characteristics of the projector that enforces or hinders it, how it aligns with test accuracy), so as to better understand what SSL projectors do and how to improve it. As explained in the Experiment section:
>
> 5.1 is a sanity check: it assesses pairwise independence minimization for different SSL methods and shows that VICReg and Barlow Twins achieve better levels of HSIC (by design).
>
> 5.2 validates the theory from section 3 and demonstrates an improvement of VICReg obtained with our better understanding of the projector as well as suggesting other paths for future improvement.
>
> 5.3 shows that the combination of a random projector with VCReg can be useful outside of SSL (while still behaving according to our theory), for example to solve ICA problems.
>
> ### Training the projector causes the independence to fail.
>
> We want to correct a potential misunderstanding: training the projector to optimize VC only (i.e., without the invariance term) indeed causes the independence to fail, which we get in the context of ICA where the invariance term is not needed (see last paragraph). In the context of SSL, where an invariance term is present, the independence improves throughout the training even when the projector is learned (Figure 1 left, for example). We claim this is because the projector optimizes the invariance criterion, and the weights move quasi independently from the VC objective.

---

### Official Review · Reviewer_UspR · 2022-10-25

**Confidence:** 2
**Correctness:** 3
**Technical Novelty And Significance:** 3
**Empirical Novelty And Significance:** 2
**Recommendation:** 5

**Clarity, Quality, Novelty And Reproducibility:**

Novelty: This study is somewhat novel.

Quality: The analysis is comprehensive.

Clarity: This paper is well-written and well-organized.

Reproducibility: The authors provide the necessary details for reproducibility. The code is not provided.

**Strength And Weaknesses:**

Strengths:

1: This paper studies the VCRed regularization in self-supervised learning algorithms. This problem is interesting and could have a high impact on the community.

2: This paper is well-written and well-organized. The analysis is comprehensive.


Weaknesses:
1. The motivation is unclear to me. Can the study on independence explain why self-supervised learning works or help improve the existing SSL methods?

2. The contributions are limited. The results in this paper do not provide enough useful intuition or knowledge for improving SSL.


**Summary Of The Paper:**

This paper studies the VCRed regularization in self-supervised learning algorithms. The authors show that VCReg enforces pairwise independence between the features of the learned representation. Detailed analysis is provided.

**Summary Of The Review:**

This paper studies the VCRed regularization in self-supervised learning algorithms. However, the motivation is not clear and the results are not interesting.

---

> ### Author Response · Authors · 2022-11-16
> **Motivation of the work and importance of the results.**
>
> We thank the reviewer for pointing out that our motivations did not transpire clearly in our original submission. We have made sure to update those in the revised manuscript by slightly tweaking the abstract and introduction reinforcing the fact that our principal motivation is to better understand the role of the projector in SSL, and that VCReg is a means for us to do so. We also suggest why pairwise independence could in fact be a favorable property in a representation in the revised conclusion, along with new experiments. See general comments for detailed answers on motivation and contribution.
>
> ### Does this work improve SSL?
>
> We also emphasize that our current theoretical understanding of the projector network in SSL is very limited (see Appendix A where we thoroughly review existing works), and thus our work takes a significant step in providing a clear understanding on the role of the projector network, with a methodology that could be generalized outside of SSL. Besides, our work improves VICReg (see experiments with modified batch norm Figure 2 top right) and suggests further improvements of the projector left for future work.
>
>
> ### Interest outside SSL.
>
> Finally, and outside of SSL, we demonstrate that VC applied to a random projector can be a simple way to enforce pairwise independence in applications where it is central, such as ICA (Section 5.3) or learning disentangled representations (not treated here) which to the best of our knowledge was never used, maybe by lack of theoretical grounding that our work now provides.

---

### Official Review · Reviewer_PbKY · 2022-10-26

**Confidence:** 3
**Correctness:** 3
**Technical Novelty And Significance:** 2
**Empirical Novelty And Significance:** 2
**Recommendation:** 6

**Clarity, Quality, Novelty And Reproducibility:**

The paper is well-written. It seems the authors expect readers have knowledge in both vision dnn and kernel independence test. I know the latter a bit more and wish the authors could have explained the vision part in more details --- the paper is comprehensible.



**Strength And Weaknesses:**

Strength: The direct connection the authors made between kernel-based independence test and the regularization seems to be quite interesting. Non-trivial amount of analysis is needed to make the connection rigorous.

Weakness:
It seems that the analysis relies on the projection is random and there was a remark on that.

Also, there was a discussion on the fully independence (Eq. 4): I guess I am a bit lost: do we have fully independence if we have pairwise independence under the specialized test (usually the answer is no? but I feel less sure because I dont have too much intuition on the kernel-based test). If we need fully independence, does that mean we need exp(d) samples, which is impossible in practice?

**Summary Of The Paper:**

The paper aims to connect an effective regularization technique used in deep learning for visions with a special kind of measuring independence technique through kernel method. This connection would show that the regularization's goal is to suppress correlation of learned representations. Then the authors proceed to validate their claim via experiments.

The central idea is quite interesting (though I doubt whether that's very new): a random projection would intuitively give you a set of good basis and if the features are not correlated along these basis, then they should not be correlated. I am aware that Andoni, Panigraphy, Valiant, and Zhang's work utilizes a similar idea (and I believe this is well known enough so that probably there are other works doing similar things). The authors are aware of some weakness associated with this approach, e.g., it needs to assume the projection is random and not learned, and it appears that there does not exist any technique to circumvent the problem.

Overall, it feels like a paper that can appear in top venues but it is at the borderline side.

**Summary Of The Review:**

1. it builds a quite interesting connection between deep learning and independence test.
2. it is around the borderline papers accepted to neurips/icml/iclr (calibration).
3. while the specific connection is new, the idea of taking advantages of random projections seems to be around for a while.
4. the projection has to be random to make the analysis work. The authors seem to be aware of the weakness.

---

> ### Author Response · Authors · 2022-11-16
> **Full independence and pairwise independence under the specialized test.**
>
> We thank the reviewer for the thorough feedback. Following the reviewer’s suggestion, we provide more details on the vision part in Appendix A of the revised manuscript, and also propose a possible path to study trained projectors in our new Conclusion. See general comments on the novelty of our work and the use of random projectors.
>
> ### Full independence and pairwise independence under the specialized test.
>
> For a random vector, mutual (or full) independence implies pairwise independence for all pairs of components but the converse is generally not true. This does not depend on the specialized test as long as the kernel used for the test is universal. As an illustration, our work shows that pairwise independence (for all pairs) is optimized by the combination of a projector with VC, but full independence is not optimized (see Plateau in Figure 2 right, and 5.3 where the projector + VCReg fails to solve nonlinear ICA for which full independence is necessary).

---

### Author Response · Authors · 2022-11-16
**General answer**

We thank the referees for their thorough reviews which have made our manuscript stronger. We ought to highlight that our work is the first principled explanation for MLP projectors, studied via VICReg, BarlowTwins and W-MSE. The projector is a crucial part of state-of-the-art SSL methods with joint embeddings, responsible at large for the achieved test accuracy on ImageNet (SSL training without projector leads to a drop of about 20 points of percentage in SimCLR or VICReg). Until now, MLP projectors were eluded by all theoretical papers on the topic, making our work a potentially important step towards understanding SSL. We added the corresponding literature review in introduction and Appendix A of the revised version. We understand that all reviewers acknowledge that a better understanding of SSL regularizations is important and that our analysis was well pursued. The main limitations of the paper that emerged among some reviewers are:


1. Limited interest in studying pairwise independence in the representation, as there is no clear relation between HSIC and downstream task performance.
2. Limitation of the theoretical analysis to random weights projectors.



### 1. What is the interest of studying pairwise independence: this is what SSL projectors combined with VCReg do. Besides, this property could be useful for linear probing and model selection.


First, the focus of our work is not to advocate for pairwise independence in representations: it simply puts in evidence and explains how this property is enforced by SSL projectors combined with VCreg. We thus feel that the actual value (or lack thereof) of enforcing pairwise independence is not our argument and should not be used as a limitation of our results; rather it could be used jointly with our findings to potentially design improved SSL methods that would refrain from enforcing this property (if found undesirable).

Generally, a conclusion of our work is that lower HSIC is beneficial until a certain point (see inflection in Figure 2 top right and new Figure 5) because it does not guarantee that useful information is retained by the encoder. Nevertheless, we believe HSIC could be useful in SSL, whose methods are usually evaluated via linear probing on downstream tasks, i.e., a linear classifier is trained on top of the representation. The following table suggests that pairwise independence makes the representation easier to probe linearly: given equivalent test accuracy on ImageNet, representation with lowest HSIC may deliver better performance on downstream tasks. Pairwise independence (hence the projector) may allow the representation to be linearly separable (as opposed to MAE [1] which does not have a projector and is not good at linear probing). In this perspective, HSIC could be used for model selection for downstream tasks at least in linear evaluation. This could also settle the question on the importance of the projector. We leave these important questions for future work.

|                      | HSIC  | ImageNet | Places205  | iNat       | CO3D       | CO3D video   |
|----------------------|-------|----------|------------|------------|------------|--------------|
| VICReg               | 0.337 | **68.0**     | 46.5 ± 0.2 | 27.6 ± 0.1 | 97.7 ± 0.1 | 98.1 ± 0.1   |
| VICreg no bn in projector | 0.239 | 67.6     | **47.1 ± 0.1** | **29.9 ± 0.2** | **98.0 ± 0.1** | **98.53 ± 0.03** |

[1] He et al., Masked Autoencoders Are Scalable Vision Learners (2021)

### 2. The analysis is limited to random projectors: this is not a strong assumption since the projector should not optimize VC.

Indeed:
- We experimentally find that learning the projector to optimize VC by removing the invariance term leads to a form of collapse (see ICA experiments). For example, a projector optimizing VC only could trivially learn Z = (X_1, 0, …, 0). In fact, random projectors lead to better HSIC, further suggesting that learning the projector is orthogonal to optimizing HSIC.
- Hence, in VICReg, it is reasonable to model the weights of the projectors as independent from the VC term, hence keeping them random for the analysis.
- Finally, learned projectors experimentally behave according to our theory (see e.g. behavior w.r.t. width and depth), which could be linked to existing results in over-parametrized models for which trained weights stay at (or near) their initial (random) values [2] (now mentioned in the conclusion).

[2] Jacot et al., Neural Tangent Kernel: Convergence and Generalization in Neural Networks (2018)

We have made sure to address those two limitations by providing novel experiments and by emphasizing throughout our submission that our analysis applies exactly only to random projectors. Nevertheless, we hope that this first step towards theoretically understanding the role of the projector can be seen as a significant progress on its own. We remain available for further discussions and would welcome any further comment from the reviewers.

---

### Author Response · Authors · 2022-11-16
**List of revisions**

Please find our revised manuscript, where we made sure to address the concerns made by the reviewers. List of notable changes:
- Emphasized the interest of our work to understand projectors in SSL in Introduction and Appendix A.
- Added comments on the relationship between HSIC and test accuracy in the Experiments section and in Appendix C.
- Added discussion and new experiments showing the usefulness of pairwise independence in representation learning for model selection and linear probing in new Conclusion.
- Added clarification and discussion on the assumption of random weights throughout the manuscript and in new Conclusion.
- Improved clarity and figures following MKGJ’s and EpBg’s suggestions.
- Added more explanation on the vision side following PbKY’s suggestion.

---

### Author Response · Authors · 2022-12-12
**End of rebuttal period**

The rebuttal period is nearly concluded, and we thank the reviewers for their time and consideration for our rebuttal. For those who did not assess the rebuttal yet, it would be very helpful to us if you could answer whether your concerns were addressed! We have put a significant effort into thoroughly addressing the feedback of each reviewer with new experiments, clarifications,  updates to the manuscript, and we believe our submission to be stronger now.

Thanks,

The authors

---

### Decision · Program_Chairs · 2023-01-20

**Decision:**

Reject

**Justification For Why Not Higher Score:**

There was a strong concern regarding the motivation and usefulness of this paper among reviewers.

**Justification For Why Not Lower Score:**

N/A

**Metareview: Summary, Strengths And Weaknesses:**

The paper studies the VCRed regularization in self-supervised learning algorithms and shows that it enforces pairwise independence between the features of the learned representation. While there are positive aspects to the paper, there was a key consistent concern among reviewers that leads me to recommend rejecting this paper: Reviewers thought that the paper did not have a sufficient motivation. The results do not seem to be surprising given prior work and it is also highly unclear how the paper leads to better SSL algorithms.